# FEATURE-BASED ONLINE BILATERAL TRADE

**Solenne Gaucher**[1]   **Martino Bernasconi**[2]   **Matteo Castiglioni**[3]   **Andrea Celli**[2]
**Vianney Perchet**[4,5]
[1] CMAP, École polytechnique, IPP   [2] Bocconi University
[3] Politecnico di Milano   [4] Criteo AI Lab   [5] Fairplay team, CREST, ENSAE, IPP
`{martino.bernasconi,andrea.celli2}@unibocconi.it`,
`matteo.castiglioni@polimi.it`, `vianney.perchet@normalesup.org`,
`solenne.gaucher@polytechnique.edu`

## ABSTRACT

Bilateral trade models the problem of facilitating trades between a seller and a buyer having private valuations for the item being sold. In the online version of the problem, the learner faces a new seller and buyer at each time step, and has to post a price for each of the two parties without any knowledge of their valuations. We consider a scenario where, at each time step, before posting prices the learner observes a context vector containing information about the features of the item for sale. The valuations of both the seller and the buyer follow an unknown linear function of the context. In this setting, the learner could leverage previous transactions in an attempt to estimate private valuations. We characterize the regret regimes of different settings, taking as a baseline the best context-dependent prices in hindsight. First, in the setting in which the learner has two-bit feedback and strong budget balance constraints, we propose an algorithm with $O(\log T)$ regret. Then, we study the same set-up with noisy valuations, providing a tight $\widetilde{O}(T^{2/3})$ regret upper bound. Finally, we show that loosening budget balance constraints allows the learner to operate under more restrictive feedback. Specifically, we show how to address the one-bit, global budget balance setting through a reduction from the two-bit, strong budget balance setup. This established a fundamental trade-off between the quality of the feedback and the strictness of the budget constraints.

## 1 INTRODUCTION

Bilateral trade models scenarios in which a seller and a buyer, both having a private valuation for a good, are interested in trading it in order to maximize their respective utilities (Vickrey, 1961; Myerson & Satterthwaite, 1983). We study the online bilateral trade problem introduced by Cesa-Bianchi et al. (2021). At each time $t$, a new seller and buyer arrive, with private valuations $s_t$ and $b_t$, respectively. The seller's valuation $s_t$ is the lowest price they are willing to accept for the item. Analogously, the buyer's valuation $b_t$ represents the highest price they are willing to pay for the item. The learner, without any knowledge about the private valuations at the current time $t$, posts two (possibly randomized) prices: $p_t$ to the seller and $q_t$ to the buyer. A trade happens when $s_t \le p_t$ and $q_t \le b_t$, so both agents agree to trade. The *gain from trade* (GFT) for a pair of prices $(p, q)$ at time $t$ is

$$\text{GFT}_t(p, q) := \mathbb{I}\{s_t \le p\}\mathbb{I}\{q \le b_t\}(b_t - s_t). \tag{1}$$

This represents the net utility gain generated by the trade. Following online bilateral trade literature (see, *e.g.,* Cesa-Bianchi et al. (2021; 2023); Bernasconi et al. (2024)), we assume the learner aims to maximize trade gain or equivalently minimize regret relative to the best hindsight policy.[1] A key challenge in online bilateral trade is the inherently limited feedback: under *two-bit feedback*, the learner receives feedback $(\mathbb{I}\{s_t \le p_t\}, \mathbb{I}\{b_t \ge q_t\})$, and under *one-bit feedback* the learner only observes whether the trade occurred, that is $\mathbb{I}\{s_t \le p_t\} \cdot \mathbb{I}\{b_t \ge q_t\}$. Both provide far less informa-

---

[1] We note that gain from trade and social welfare (*i.e.,* the sum of the utilities of both agents if the trade occurs, or the seller's utility if it does not) result in the same notion of regret.

tion than traditional bandit feedback, since the learner cannot even reconstruct the gain from trade received for the prices it posted.

In the standard model of bilateral trade, the platform lacks information about the seller, buyer, or the item being sold. However, this scenario is unrealistic in practice, where some information is usually available to the learner. In contexts like online marketplaces, where products are often differentiated, the learner can observe product features upfront and base pricing decisions on them, using past trade data to estimate feature values and inform future pricing.

We introduce the *feature-based* online bilateral trade problem, in which the learner observes a feature vector $x_t \in \mathbb{R}^d$ before posting prices for round $t$. Building on the traditional feature-based dynamic pricing framework (see, *e.g.,* Cohen et al. (2020); Javanmard (2017); Javanmard & Nazerzadeh (2019); Keskin & Zeevi (2014); Xu & Wang (2021)), we study the setting in which private valuations are of the form $x_t^\top \theta + \xi_t$, where $\theta \in \mathbb{R}^d$ is an unknown vector denoting the importance of each feature, and $\xi_t$ is an i.i.d. noise term. Following the set-up of Cohen et al. (2020), feature vectors $x_t$ are chosen adversarially by nature. This ensures that our solution is robust to scenarios where features are correlated and where the set of relevant features evolves over time.

## 1.1 OUR CONTRIBUTIONS

In this paper, we introduce the feature-based online bilateral trade model, and characterize the regret for various scenarios with adversarially generated feature vectors.

First, we focus on the scenario in which the learner has two-bit feedback and strong budget balance constraints (the learner has to set the same price to the seller and to the buyer, *i.e.,* $p_t = q_t$ at each round $t$). As a preliminary warm-up, we begin by considering the deterministic setup where, at each $t$, the seller's valuation is $s_t = x_t^\top \theta^s$ and the buyer's valuation is $b_t = x_t^\top \theta^b$. We show that in this case, it is possible to adapt techniques proposed by Cohen et al. (2020) in the context of feature-based dynamic pricing. The main difference here is that we need to maintain separate ellipsoidal uncertainty sets for the seller and the buyer, respectively. Our analysis yields a regret of order $O(\log T)$ (Proposition 1).

**Two-bit, noisy valuations.** Then, we consider the case of noisy valuations in which $s_t = x_t^\top \theta^s + \xi_t^s$ and $b_t = x_t^\top \theta^b + \xi_t^b$, where $\xi_t^s$ and $\xi_t^b$ are i.i.d. noise terms independent from $x_t$, with bounded support and densities. We start by decomposing the expected gain-from-trade into components that can be individually estimated using two-bit feedback (Lemma 1). Then, we describe an explore-*or*-commit (EOC) algorithm with regret $\widetilde{O}(T^{3/4})$ (Theorem 1) (this algorithm will be employed in the subsequent reduction to the one-bit setting). Finally, we devise a SCOUTING BANDITS WITH INFORMATION POOLING algorithm, which makes a more efficient use of the information collected during the learning phase. This algorithms achieves a regret $\widetilde{O}(T^{2/3})$(Theorem 2), which is minimax optimal up to poly-logarithmic factors and dependence on $d$. Indeed, there exists a matching $\Omega(T^{2/3})$ lower bound for the stochastic bilateral trade problem without features in the case of independent valuations with bounded densities and support Cesa-Bianchi et al. (2021).

**One-bit reduction.** Finally, we provide a general black-box reduction that demonstrates how the difficulty of maintaining a per-round budget balance can be traded for the ability to operate under more demanding feedback conditions. In particular, given a two-bit explore-or-commit algorithm guaranteeing strong budget balance and sublinear regret, we show that it is possible to construct a no-regret algorithm that works under one-bit feedback, and is global budget balanced (Theorem 3). The one-bit regret guarantees are dependent on a natural measure of the social welfare generated by the market.

## 1.2 RELATED WORKS

**Bilateral trade.** In the offline setting, Myerson & Satterthwaite (1983) showed the existence of instances where a fully efficient mechanism that satisfies incentive compatibility, individual rationality, and budget balance does not exist. Subsequent research focused on finding approximately efficient mechanisms in the Bayesian setting Blumrosen & Dobzinski (2014); Kang et al. (2022); McAfee (2008); Blumrosen & Mizrahi (2016); Brustle et al. (2017); Deng et al. (2022); Fei (2022).

**Online bilateral trade.** Cesa-Bianchi et al. Cesa-Bianchi et al. (2021); Cesa-Bianchi et al. (2024) study the case in which valuations are drawn i.i.d. from some fixed unknown distribution and the learner has to enforce strong budget balance. They provide sublinear regret guarantees in the full-feedback setting, and under partial feedback when valuations are i.i.d. samples from a smooth distribution, independently for the seller and the buyer. If the learner is only required to enforce *weak budget balance* (*i.e.,* $p_t \leq q_t$ for each $t$), then Azar et al. (2022) provide an algorithm achieving a tight sublinear 2-regret when the sequence of valuation is generated by an oblivious adversary. Cesa-Bianchi et al. (2023) show that sublinear regret can be achieved beyond the i.i.d. stochastic setting under a $\sigma$-smooth adversary model. Bernasconi et al. (2024) show that sublinear regret can be achieved in the fully adversarial setting if the learner enforces global budget balance constraints (*i.e.,* the constraint has to hold over the entire time horizon). A related setting to online bilateral trade is online brokerage (Bolić et al., 2024). This model has some key structural differences from the standard online bilateral trade framework by Cesa-Bianchi et al. (2021). For instance, traders can act as both buyers and sellers depending on market conditions, and their valuations are i.i.d. from the same unknown distribution. Bachoc et al. (2024) recently studied a contextual brokerage model where traders' valuations are zero-mean perturbations of a market price that is linear in the feature vector.

**Feature-based pricing.** The one-dimensional version of the problem was introduced by Kleinberg & Leighton (2003), while Amin et al. (2014) introduced the problem in the contextual, multi-dimensional set-up under i.i.d. contexts. Cohen et al. (2020) study a model with adversarial contexts for which they provide $\widetilde{O}(d^2 \log T)$ regret guarantees in the noiseless setting, improving over the $\widetilde{O}(\sqrt{T})$ guarantees obtainable with general-purpose contextual bandits algorithms Agarwal et al. (2014). Moreover, they provide $\widetilde{O}(T^{2/3})$ regret guarantees for scenarios with adversarial contexts and additive noise generated from a known sub-Gaussian distribution. Further improvements in the achievable rates were later provided by Lobel et al. (2018); Leme & Schneider (2018); Liu et al. (2021). Further instantiations of the feature-based pricing model can be found in the survey of Den Boer (2015).

## 2   PRELIMINARIES

For any $k \geq 1$, we compactly denote the set $\{1, 2, \ldots, k\}$ as $[\![k]\!]$. We denote by $\mathrm{LJ}(H)$ the Löwner-John ellipsoid of a convex body $H$ (*i.e.,* the minimal volume ellipsoid that contains $H$). For a positive definite matrix $\mathbf{M} \in \mathbb{R}^{d \times d}$ and $x \in \mathbb{R}^d$, we denote $\sqrt{x^\top \mathbf{M} x}$ by $\|x\|_{\mathbf{M}}$. In our proof sketches, we write $A_T \lesssim B_T$ if there exists a problem-dependent constant $c$ such that $A_T \leq c B_T$.

**Learning protocol.** At each round $t$, a new buyer and a seller arrive, characterized by private valuations $b_t$ and $s_t$, respectively. After observing a feature vector $x_t \in \mathbb{R}^d$, the learner posts two prices: $p_t$ to the seller and $q_t$ to the buyer. The trade happens if both agents accept the proposed prices, that is $s_t \leq p_t$ and $q_t \leq b_t$. When the trade happens the learner is awarded the gain from trade $\mathrm{GFT}_t(p_t, q_t)$, defined as per Equation (1). Feature vectors $x_t$ are generated by an oblivious adversary.

**Valuations.** We consider two models to describe how features impact on the seller's and buyer's valuations. In the first setting, valuations are a **deterministic** function of the context. At each round $t \in [\![T]\!]$, the valuations of the seller and of the buyer are given by

$$s_t = x_t^\top \theta^s \quad \text{and} \quad b_t = x_t^\top \theta^b, \tag{2}$$

respectively, where $\theta^s, \theta^b \in \mathbb{R}^d$ are unknown to the learner. We assume that both the contexts $x_t$ and the parameters $\theta^b$ and $\theta^s$ are bounded.

**Assumption 1.** *There exist $A, B \geq 0$ such that $\max(\|\theta^s\|, \|\theta^b\|) \leq A$, and $\|x_t\| \leq B$.*

In the second setting, we consider **noisy** valuations. At each round $t \in [\![T]\!]$, the valuations are given by

$$b_t = x_t^\top \theta^b + \xi_t^b \quad \text{and} \quad s_t = x_t^\top \theta^s + \xi_t^s, \tag{3}$$

where $\xi_t^b$ and $\xi_t^s$ are i.i.d. centered noise terms independent from the context $x_t$, with densities $f^b$ and $f^s$, respectively. We denote by $F^s$ the c.d.f. of $\xi_t^s$, and by $D^b$ the demand function of $\xi_t^b$: for all $x \in \mathbb{R}$, $D^b(x) := \mathbb{P}\left(\xi_t^b \geq x\right)$. We make the following assumption on the densities of $\xi_t^s$ and $\xi_t^b$.

---

**Algorithm 1** ELLIPSOIDPRICING FOR BILATERAL TRADE (EP-BT)

---

1: **Input** parameter $\epsilon > 0$, bound $A$.
2: **Initialize** $K_1^s, K_1^b \leftarrow d$-dimensional ball of radius $A$, $E_1^s = \text{LJ}(K_1^s)$ and $E_1^b = \text{LJ}(K_1^b)$.
3: **for** $t \in [\![T]\!]$ **do**
4:     Set $\underline{s_t} = \min_{\theta \in E_t^s} x_t^\top \theta$ and $\overline{s_t} = \max_{\theta \in E_t^s} x_t^\top \theta$ and compute $(\underline{b_t}, \overline{b_t})$ analogously
5:     **if** $\overline{s_t} < \underline{b_t}$ **then** post price $p_t = (\overline{s_t} + \underline{b_t})/2$
6:     **else if** $\overline{s_t} - \underline{s_t} \geq \epsilon$ **then**
7:         Post price $p_t = (\overline{s_t} + \underline{s_t})/2$
8:         Update $K_{t+1}^s$ according to the seller's feedback, set $E_{t+1}^s = \text{LJ}(K_{t+1}^s)$
9:     **else if** $\overline{b_t} - \underline{b_t} \geq \epsilon$ **then**
10:       Post price $p_t = (\overline{b_t} + \underline{b_t})/2$
11:       Update $K_{t+1}^b$ according to the seller's feedback, set $E_{t+1}^b = \text{LJ}(K_{t+1}^b)$
12:     **else** post price $p_t = (\overline{s_t} + \underline{b_t})/2$

---

**Assumption 2.** *Both $\xi_t^s$ and $\xi_t^b$ are supported in $[-C, C]$, and have densities bounded by $L$.*

We observe that under Assumption 1 and 2, the valuations of the seller and the buyer are bounded in $[-P, P]$, where $P := C + AB$.[2] The bounded densities assumption is standard in repeated bilateral trade problems with stochastic valuations, and there exists linear regret lower bounds for the case in which this assumption is lifted (Cesa-Bianchi et al., 2024).

**Feedback models.** After posting prices $(p_t, q_t)$, the learner does not observe directly the GFT or the valuations, but receives some feedback $z_t$ about the transaction. We focus on two feedback models: (i) *Two-bit feedback*, where both the buyer and the seller reveal their willingness to accept the prices offered by the learner (*i.e.*, $z_t$ is composed by the two bits $(\mathbb{I}\{s_t \leq p_t\}, \mathbb{I}\{q_t \leq b_t\})$); (ii) *One-bit feedback*: the learner only observes whether the trade happened or not (*i.e.*, $z_t = \mathbb{I}\{s_t \leq p_t\} \cdot \mathbb{I}\{q_t \leq b_t\}$). Both feedback models have the desirable property of revealing minimal information about agent's private valuations.

**Regret.** Our objective is to develop dynamic policies that perform well in terms of minimizing regret. In the deterministic setting, the worst-case regret for learning algorithm $\mathcal{A}$ is defined as

$$R_T(\mathcal{A}) := \max_{(\theta^s, \theta^b) \in [-A, A], \boldsymbol{x} \in [-C, C]^T} \sum_{t \in [\![T]\!]} \left( [x_t^\top (\theta^b - \theta^s)]^+ - \text{GFT}_t(p_t, q_t) \right).$$

In the noisy model, we compare against a benchmark that maximizes the expected gain from trade. Let $\text{EGFT}(x, p, q) := \mathbb{E}[\text{GFT}_t(p, q) \mid x_t = x]$. The pseudo-regret for the noisy setting is defined as

$$R_T(\mathcal{A}) := \sum_{t \in [\![T]\!]} \max_{\substack{(p, q) \in [-P, P]^2 \\ p \leq q}} \text{EGFT}(x_t, p, q) - \sum_{t \in [\![T]\!]} \text{EGFT}(x_t, p_t, q_t).$$

A property that directly follows by definition is that, for any feature vector, there exists a pair of identical prices that maximize the expected gain from trade. To simplify the notation, we omit the argument $q$ of $\text{GFT}_t$ and $\text{EGFT}$ if the same price is posted to both agents.

## 3 WARM-UP: TWO-BIT FEEDBACK, STRONG BUDGET BALANCE, NO NOISE

In this section, we consider the simplest set-up in terms of the information available to the learner, by focusing on the setting with two-bit feedback and deterministic valuations following Equation (2). We adapt the ELLIPSOIDPRICING algorithm, originally proposed by Cohen et al. (2020) for feature-based dynamic pricing, to our bilateral trade problem. The key distinction lies in managing separate uncertainty sets for the seller and the buyer, while carefully setting prices consistent with both estimations. In this set-up, we can update these uncertainty sets separately. Algorithm 1 describes the main step of ELLIPSOIDPRICING FOR BILATERAL TRADE (EP-BT).

**Proposition 1.** *Assume that the seller's and buyer's valuations follow Equation (2). Under Assumption 1, Algorithm 1 with $\epsilon = ABd^2/T$ has regret bounded by $R_T \leq 10ABd^2 \log\left(20(d+1)Td^{-2}\right)$.*

---

[2]For simplicity, we work within $[-P, P]$. If non-negative prices are required, a translation suffices.

---

**Algorithm 2** ESTIMATION SUBROUTINES

---

1: **procedure** EST-PAR($\mathcal{T}^{\text{par}}$)                                                  ▷Update estimate of parameters $\theta^b, \theta^s$
2:     Draw price $p_t \sim \mathcal{U}([-P, P])$ and post $(p_t, p_t)$
3:     $\mathbf{V} \leftarrow \sum_{l \in \mathcal{T}^{\text{par}}} x_l x_l^\top + \mathbf{I}_d$
4:     $\widehat{\theta}^s \leftarrow 2P\mathbf{V}^{-1} \sum_{l \in \mathcal{T}^{\text{par}}} \left( \mathbb{I}\{p_l \leq s_l\} - \frac{1}{2} \right) x_l$
5:     $\widehat{\theta}^b \leftarrow 2P\mathbf{V}^{-1} \sum_{l \in \mathcal{T}^{\text{par}}} \left( \mathbb{I}\{p_l \leq b_l\} - \frac{1}{2} \right) x_l$

---

6: **procedure** EST-INT($\mathcal{T}^{\text{int}}$)                                                  ▷Update estimate of integrals $I, J$
7:     Draw price $p_t \sim \mathcal{U}([-P, P])$ and post $(p_t, p_t)$
8:     **for** $k \in \mathcal{K}$ **do**
9:         $\widehat{J}^k \leftarrow \frac{2P}{|\mathcal{T}^{\text{int}}|} \sum_{l \in \mathcal{T}^{\text{int}}} \mathbb{I} \left\{ s_l \leq p_l \leq k\epsilon + x_l^\top \widehat{\theta}^s \right\}$.
10:        $\widehat{I}^k \leftarrow \frac{2P}{|\mathcal{T}^{\text{int}}|} \sum_{l \in \mathcal{T}^{\text{int}}} \mathbb{I} \left\{ k\epsilon + x_l^\top \widehat{\theta}^b \leq p_l \leq b_l \right\}$

---

11: **procedure** EST-F($\mathcal{T}_k^{\text{F}}$, grid point $k \in \mathcal{K}$)                    ▷Update estimate of $F^s$
12:     Post price $p_t = x_t^\top \widehat{\theta}^s + k\epsilon$. Then update $\widehat{F}^k \leftarrow \sum_{l \in \mathcal{T}_k^{\text{F}}} \mathbb{I}\{s_l \leq p_l\} / |\mathcal{T}_k^{\text{F}}|$

---

13: **procedure** EST-D($\mathcal{T}_k^{\text{D}}$, grid point $k \in \mathcal{K}$)                    ▷Update estimate of $D^b$
14:     Post price $p_t = x_t^\top \widehat{\theta}^b + k\epsilon$. Then update $\widehat{D}^k \leftarrow \sum_{l \in \mathcal{T}_k^{\text{D}}} \mathbb{I}\{b_l \geq p_l\} / |\mathcal{T}_k^{\text{D}}|$

---

We observe that the algorithm explores only whenever the estimations are not precise enough. In the proof (Appendix B), we show that this happens in at most $O(\log T)$ rounds.

## 4 NOISY VALUATIONS WITH TWO-BIT FEEDBACK

In this section, we consider bilateral trade with two-bit feedback and noisy valuations. We introduce key components that we exploit in the analysis, before integrating them into an explore-*or*-commit framework which ensures a regret $\widetilde{O}(T^{3/4})$. Finally, we show a careful re-design of the last phase of the algorithm yields a tight regret upper bound of $\widetilde{O}(T^{2/3})$.

We rely on the following decomposition lemma, which extends Lemma 1 of Cesa-Bianchi et al. (2024) to contextual settings. The proof can be found in Appendix E.1.

**Lemma 1.** *Under Assumptions 1 and 2, the expected gain from trade for $x$ at price $p$ is given by*

$$\text{EGFT}(x, p) = I(\delta^b) F^s(\delta^s) + J(\delta^s) D^b(\delta^b), \tag{4}$$

*where $\delta^s := p - x_t^\top \theta^s$, $\delta^b := p - x_t^\top \theta^b$, $I(\delta) := \int_\delta^C D^b(u) du$, and $J(\delta) := \int_{-C}^\delta F^s(u) du$.*

Lemma 1 emphasizes that the expected gain from trade for a price $p$ depends on the pair of price increments $(\delta^s, \delta^b)$, or equivalently, on the pair $(\delta^s, \Delta_t)$, where the difference in average valuations $\Delta_t$ is defined as $\Delta_t := x_t^\top(\theta^b - \theta^s)$. This shows that items with the same $\Delta_t$ have the same optimal seller's price increment $\delta_t^*$: if $p = x_t^\top \theta^s + \delta_t^*$ maximizes $\text{EGFT}(x_t, \cdot)$, and if $x_{t'}^\top(\theta^b - \theta^s) = x_t^\top(\theta^b - \theta^s)$, then $p' = x_{t'}^\top \theta^s + \delta_t^*$ maximizes $\text{EGFT}(x_{t'}, \cdot)$. However, knowing the optimal increment $\delta_t^*$ for a specific $\Delta_t$ is not sufficient to determine the optimal increment for a different $\Delta_t$ (see Appendix F). Thus, finding the optimal price increment across various $\Delta_t$ values might require us to precise estimate the reward function over a broad range of increments. This is a notable departure from the non-contextual stochastic bilateral trade problem discussed in Cesa-Bianchi et al. (2024), where precise estimation of the reward function around the optimal price suffices. Since accurately estimating functions such as $F^s$, $D^b$, $I$, and $J$ across a wide range of arguments complicates the problem, the regret might be higher than the $\widetilde{O}(T^{2/3})$ rate achieved in Cesa-Bianchi et al. (2024). Surprisingly, we show that with careful coordination of the various estimation procedures, the optimal rate for the non-contextual case can still be attained.

---

**Algorithm 3** EXPLORE-OR-COMMIT FOR BILATERAL TRADE (EOC-BT)

1: **Input**: parameter $\mu$, length of estimation phases $\mathrm{T^{int}}$, $\mathrm{T^{FD}}$, discretization error $\epsilon$, confidence $\delta$.
2: **Initialize**: $K = \lceil 2P/\epsilon \rceil + 3$, $\mathcal{K} \leftarrow [\![-K, K]\!]$, $\mathcal{T}^{\mathrm{par}} = \mathcal{T}^{\mathrm{int}} = \mathcal{T}_k^{\mathrm{F}} = \mathcal{T}_k^{\mathrm{D}} = \varnothing$ for all $k \in \mathcal{K}$, $\mathbf{V} = \mathbf{I}_d$.
3: **while** $t \leq T$ **do**
4:    **if** $\|x_t\|_{\mathbf{V}^{-1}} > \mu$ **then** $\mathcal{T}^{\mathrm{par}} \leftarrow \mathcal{T}^{\mathrm{par}} \cup \{t\}$, EST-PAR($\mathcal{T}^{\mathrm{par}}$)
5:    **else if** $|\mathcal{T}^{\mathrm{int}}| < \mathrm{T^{int}}$ **then** $\mathcal{T}^{\mathrm{int}} \leftarrow \mathcal{T}^{\mathrm{int}} \cup \{t\}$, EST-INT($\mathcal{T}^{\mathrm{int}}$)
6:    **else if** for some $k \in \mathcal{K}$, $|\mathcal{T}_k^{\mathrm{F}}| < \mathrm{T^{FD}}$ or $|\mathcal{T}_k^{\mathrm{D}}| < \mathrm{T^{FD}}$ **then**
7:      **if** $|\mathcal{T}_k^{\mathrm{F}}| < \mathrm{T^{FD}}$ **then** $\mathcal{T}_k^{\mathrm{F}} \leftarrow \mathcal{T}_k^{\mathrm{F}} \cup \{t\}$, EST-F($\mathcal{T}_k^{\mathrm{F}}, k$)
8:      **else if** $|\mathcal{T}_k^{\mathrm{D}}| < \mathrm{T^{FD}}$ **then** $\mathcal{T}_k^{\mathrm{D}} \leftarrow \mathcal{T}_k^{\mathrm{D}} \cup \{t\}$, EST-D($\mathcal{T}_k^{\mathrm{D}}, k$)
9:    **else**                                          ▷Exploitation phase
10:      $\mathcal{A}_t \leftarrow \left\{ (k, k') \in \mathcal{K}^2 : k' = \left\lfloor \left( k\epsilon - x_t^\top \left( \widehat{\theta}_t^b - \widehat{\theta}_t^s \right) \right) \epsilon^{-1} \right\rfloor \right\}$
11:      Choose $(k_t, k_t') \in \arg\max_{(k,k') \in \mathcal{A}_t} \widehat{F}^k \widehat{I}^{k'} + \widehat{D}^{k'} \widehat{J}^k$ and post price $p_t = x_t^\top \widehat{\theta}^s + k_t \epsilon$

---

## 4.1 BUILDING BLOCKS: SUBROUTINES FOR LEARNING PARAMETERS

We first present the sub-routines to estimate the parameters $\theta^s$ and $\theta^b$, and the functions $F^s$, $D^b$, $I$ and $J$ used in the EGFT decomposition of Lemma 1. These sub-routines serve as the base components for the subsequent algorithms.

The first sub-routine, EST-PAR, estimates the parameters $\theta^s$ and $\theta^b$. It uses the fact that, if $p_t \sim \mathcal{U}([-P, P])$, then $2P(\mathbb{I}\{p_t \leq s_t\} - 1/2)$ is an unbiased estimate of $x_t^\top \theta^s$ (similarly for $\theta^b$). Then, we can rely on classical results to estimate the parameters (see Lattimore & Szepesvári (2020, Chapter 19)). The second sub-routine, EST-INT, estimates the integrals $I(\delta)$ and $J(\delta)$ over a grid of price increments $\{k\epsilon : k \in \mathcal{K}\}$. The third and fourth sub-routines, EST-F and EST-D, provide estimates of the c.d.f. $F^s$ and the demand function $D^b$ at $k\epsilon$ for a given increment level $k \in \mathcal{K}$, respectively.

## 4.2 EXPLORE-OR-COMMIT FRAMEWORK

First, we consider a natural Explore-*Or*-Commit (EOC) algorithm which uses each round *either* to compute estimates for the terms in Equation (4), *or* to greedily play the empirical best action. This strategy is described in Algorithm 3. Our algorithm is not a standard Explore-Then-Commit algorithm. Indeed, it resorts to the estimation subroutine EST-PAR (Line 4) whenever the estimation is not "good enough" and not only in the first rounds. The other estimation subroutines are executed until a certain number of updates is reached. We show that the number of necessary exploration rounds is upper bounded by $\widetilde{O}(T^{3/4})$. This will be useful in section Section 5 to handle one-bit feedback. Finally, in Lines 10 and 11 the algorithm selects the best price increments according to the current estimates (*i.e.*, "commit" rounds), and posts price $p_t$ build accordingly to both agents. Theorem Theorem 1 bounds the regret of EOC-BT. Its proof is postponed to Appendix C, where we provide explicit bounds on $\widetilde{T}$ and $\widetilde{C}$.

**Theorem 1.** *Set* $\epsilon = \left( \frac{\log(T)}{T} \right)^{1/4}$, $\delta = \left( T(74 + 32P\epsilon^{-1}) \right)^{-1}$, $\mu = \epsilon \left( P\sqrt{d \log\left( \frac{1 + B^2 T}{\delta} \right)} + A \right)^{-1}$, $\mathrm{T^{int}} = \lceil \frac{8P^2 \log(1/\delta)}{\epsilon^2} \rceil$, *and* $\mathrm{T^{FD}} = \lceil \frac{2 \log(1/\delta)}{\epsilon^2} \rceil$. *Then, under Assumptions 1 and 2, Algorithm 3 verifies*

$$R_T \leq \widetilde{C} T^{3/4} \log(T)^{1/4}$$

*with probability at least* $1 - T^{-1}$ *when* $T \geq \widetilde{T}$, *where* $\widetilde{T}$ *and* $\widetilde{C}$ *are constants depending respectively on* $A$, $B$, *and* $C$, *and on* $A$, $B$, $C$, *and* $d$.

*Sketch of the proof.* For some $\epsilon > 0$ to be chosen later, let $\mathcal{K} = [\![-P/\epsilon, P/\epsilon]\!]$. Using standard arguments Abbasi-Yadkori et al. (2011); Carpentier et al. (2020), we can show that $\widetilde{O}(\epsilon^{-2})$ samples are sufficient for EST-PAR to ensure that, with high probability, for all $t \notin \mathcal{T}^{\mathrm{par}}$ the errors $|x_t^\top (\theta^s - \widehat{\theta}_t^s)|$ and $|x_t^\top (\theta^b - \widehat{\theta}_t^b)|$ are smaller than $\epsilon$. When this happens, classical concentration arguments ensure that $\widetilde{O}(\epsilon^{-2})$ samples are sufficient for EST-INT to estimate $I(k\epsilon)$ and $J(k\epsilon)$ with precision

---

**Algorithm 4** SCOUTING BANDIT WITH INFORMATION POOLING (SBIP)

---

1: **Input**: parameter $\mu > 0$, length of scouting phase $\mathrm{T}^{\text{int}}$, discretization size $\epsilon$, confidence $\delta > 0$.
2: **Initialize**: $\mathcal{T}^{\text{par}} = \mathcal{T}^{\text{int}} = \varnothing$, $\mathbf{V} = \mathbf{I}_d$, $K = \lceil 2P/\epsilon \rceil + 3$, $\mathcal{K} = [\![-K, K]\!]$, $\widetilde{\epsilon} = (12PL + 7)\epsilon$,
3: $N^{k,s} = N^{k,b} = \widehat{F}^k = \widehat{D}^k = 0$ for all $k \in \mathcal{K}$. Let $\beta : \mathbb{N} \ni n \mapsto \sqrt{2\log(\delta^{-1})/n}$
4: **while** $t \leq T$ **do**
5:     **if** $\|x_t\|_{\mathbf{V}^{-1}} > \mu$ **then**
6:         $\mathcal{T}^{\text{par}} \leftarrow \mathcal{T}^{\text{par}} \cup \{t\}$, EST-PAR$(\mathcal{T}^{\text{par}})$
7:     **else if** $|\mathcal{T}^{\text{int}}| < \mathrm{T}^{\text{int}}$ **then**
8:         $\mathcal{T}^{\text{int}} \leftarrow \mathcal{T}^{\text{int}} \cup \{t\}$, EST-INT$(\mathcal{T}^{\text{int}})$
9:     **else**                               ▷Run Successive Elimination
10:         **for** $(k, k') \in \mathcal{A}_t := \left\{ (k, k') \in \mathcal{K}^2 : k' = \left\lfloor \left( x_t^\top \left( \widehat{\theta}_t^s - \widehat{\theta}_t^b \right) + k\epsilon \right) \epsilon^{-1} \right\rfloor \right\}$ **do**
11:             $\text{UCB}_t(k, k') \leftarrow \widehat{I}^{k'} \widehat{F}_t^k + \widehat{J}^k \widehat{D}_t^{k'} + \widetilde{\epsilon} + 2P \left( \beta(N^{k,s}) + \beta(N^{k',b}) \right)$
12:             $\text{LCB}_t(k, k') \leftarrow \widehat{I}^{k'} \widehat{F}_t^k + \widehat{J}^k \widehat{D}_t^{k'} - \widetilde{\epsilon} - 2P \left( \beta(N^{k,s}) + \beta(N^{k',b}) \right)$
13:         $\mathcal{K}_t := \left\{ (k, k') \in \mathcal{A}_t : \text{UCB}_t(k, k') \geq \max_{(l, l') \in \mathcal{A}_t} \text{LCB}_t(l, l') \right\}$
14:         Choose $(k_t, k_t') \in \arg\min_{(k, k') \in \mathcal{K}_t} \min\{N^{k,s}, N^{k',b}\}$ and post price $p_t = x_t^\top \widehat{\theta}_t^s + k_t \epsilon$
15:         Observe feedback and update:

$$\widehat{F}^{k_t} \leftarrow \frac{N^{k_t,s} \widehat{F}^{k_t} + \mathbb{I}\{s_t \leq p_t\}}{N^{k_t,s} + 1} \quad \text{and} \quad \widehat{D}^{k_t'} \leftarrow \frac{N^{k_t',b} \widehat{D}^{k_t'} + \mathbb{I}\{b_t \geq p_t\}}{N^{k_t',b} + 1},$$

16:         $N^{k_t,s} \leftarrow N^{k_t,s} + 1 \quad \text{and} \quad N^{k_t',b} \leftarrow N^{k_t',b} + 1$

---

$\epsilon$ uniformly for $k \in \mathcal{K}$. By contrast, $\widetilde{O}(|\mathcal{K}|\epsilon^{-2})$ samples are necessary for EST-F and EST-F to estimate $F^s(k\epsilon)$ and $D^b(k\epsilon)$ with precision $\epsilon$ uniformly for $k \in \mathcal{K}$. Thus, with high probability, after $\widetilde{O}(|\mathcal{K}|\epsilon^{-2})$ rounds of exploration, the expected gain from trade for prices $x_t^\top \widehat{\theta}_t^s + k\epsilon$ is estimated with precision $O(\epsilon)$ uniformly for $k \in \mathcal{K}$. Now, Assumption 2 ensures that the reward function is Lipschitz-continuous, so the discretization error is also $O(\epsilon)$. By choosing $\epsilon = T^{-1/4}$, we balance the estimation and discretization errors with the regret of the exploration phase, thereby obtaining a regret $\widetilde{O}(T^{3/4})$. $\qquad\qquad\qquad\qquad\qquad\qquad\qquad\qquad\qquad\qquad\qquad\qquad\qquad\qquad\square$

### 4.3 CLOSING THE GAP: SCOUTING BANDITS WITH INFORMATION POOLING

The $\widetilde{O}(T^{3/4})$ regret of the EOC strategy is mainly due to the cost of estimating the c.d.f. $F^s$ and demand function $D^b$. To bypass this costly estimation, an alternative approach is to first estimate $\theta^b$, $\theta^s$, $I$, and $J$, then run independent scouting bandit algorithms (as described in Cesa-Bianchi et al. (2024)) for each (rounded) value of $\Delta_t = x_t^\top (\theta^b - \theta^s)$. As shown in Appendix H, this strategy also achieves $\widetilde{O}(T^{3/4})$ regret. This regret rate is higher than the rate $\widetilde{O}(T^{2/3})$ achieved in non-contextual stochastic bilateral trade. Surprisingly, by combining the strengths of both strategies, we achieve a regret of $\widetilde{O}(T^{2/3})$. This reveals that the contextual stochastic bilateral trade problem is no more difficult, in terms of regret, than its non-contextual counterpart, although the comparison is against a stronger benchmark.

To achieve this optimal regret rate, we design a scouting bandit strategy with *information pooling* across different values of $\Delta_t$, presented in Algorithm 4. Information pooling is related to *cross-learning*, developed for bandits with graph feedback Balseiro et al. (2019). In particular, after having estimated $x_t^\top \theta^s$, $x_t^\top \theta^b$, $I$ and $J$, we run a successive elimination algorithm for each value of $\Delta_t$. Then, to estimate the reward of a price corresponding to increments $\delta^s$ and $\delta^b$, we use the feedback from rounds where these increments have been selected *across all values of* $\Delta_t$.

**Theorem 2.** *Under Assumptions 1 and 2, Algorithm 4 with* $\epsilon = (d^2 \log(T)^2/T)^{1/3}$, $\delta = \left((38 + \frac{16P}{\epsilon})(T+1)^2\right)^{-1}$, $\mu = \epsilon \left(P\sqrt{d\log\left(\frac{1+B^2T}{\delta}\right)} + A\right)^{-1}$, *and* $\mathrm{T}^{\text{int}} = \lceil \frac{8P^2 \log(1/\delta)}{\epsilon^2} \rceil$ *verifies*

$$R_T \leq \widetilde{C}(d\log(T)T)^{2/3}$$

*with probability at least* $1 - 1/T$ *when* $T \geq \widetilde{T}$, *where* $\widetilde{T}$ *and* $\widetilde{C}$ *are constants depending respectively on $A$, $B$, and $C$, and on $A$, $B$, $C$, and $d$.*

The regret rate of Algorithm 4 matches the lower bound $\Omega(T^{2/3})$ established by Cesa-Bianchi et al. (2021) for the simpler problem of non-contextual stochastic bilateral trade under similar assumptions (i.e., independent valuations with bounded densities and support). Thus, aside from the constant $\widetilde{C}$, the dependence on the dimension $d$, and logarithmic factors, Algorithm 4 achieves minimax-optimal regret. We outline the proof of this result below, with the full formal proof as well as bounds on $\widetilde{T}$ and $\widetilde{C}$ deferred to Appendix D.

*Sketch of the proof.* Let $r(x, p) \coloneqq \max_{p'} \text{EGFT}(x, p') - \text{EGFT}(x, p)$, and $\text{T}^{\text{par}} = |\mathcal{T}^{\text{par}}|$. The regret is bounded by $2P\text{T}^{\text{par}} + 2P\text{T}^{\text{int}} + \sum_{t \notin \mathcal{T}^{\text{par}} \cup \mathcal{T}^{\text{int}}} r(x_t, p_t)$. The elliptical potential Lemma (Carpentier et al., 2020) implies that $\text{T}^{\text{par}} \lesssim d^2 \log(T)^2 \epsilon^{-2}$, so $\text{T}^{\text{par}} + \text{T}^{\text{int}} \lesssim d^2 \log(T)^2 \epsilon^{-2}$. By using concentration arguments and exploiting the Lipschitz continuity of the reward, we can show that, for all $t \notin \mathcal{T}^{\text{par}} \cup \mathcal{T}^{\text{int}}$ and $(k, k') \in \mathcal{A}_t$, $[\text{LCB}_t(k, k'), \text{UCB}_t(k, k')]$ is a valid confidence interval for $\text{EGFT}(x_t, x_t^\top \widehat{\theta}^s + k\epsilon)$. Now, we want to show that $r(x_t, p_t) \lesssim \widetilde{O}\big(\sqrt{1/N^{k_t,s}} + \sqrt{1/N^{k'_t,b}} + \epsilon\big)$. To prove this claim, we define $(k_t^*, k_t^{'*}) \in \arg\max_{(k,k') \in \mathcal{A}_t} \text{EGFT}(x_t, x_t^\top \widehat{\theta}^s + k\epsilon)$. The Lipschitz-continuity of the reward ensures that the discretization error is of order $O(\epsilon)$. Moreover, $[\text{LCB}_t(k_t, k'_t), \text{UCB}_t(k_t, k'_t)]$ and $[\text{LCB}_t(k_t^*, k_t^{'*}), \text{UCB}_t(k_t^*, k_t^{'*})]$ are confidence intervals for $\text{EGFT}(x_t, p_t)$ and $\text{EGFT}(x_t, x_t^\top \widehat{\theta}^s + k_t^*\epsilon)$, respectively, and $\text{LCB}_t(k_t^*, k_t^{'*}) \leq \text{UCB}_t(k_t, k'_t)$ by Line 13. Then, it holds that $\text{EGFT}(x_t, x_t^\top \widehat{\theta}^s + k_t^*\epsilon) - \text{EGFT}(x_t, p_t)$ is of order $\text{UCB}_t(k_t, k'_t) - \text{LCB}_t(k_t, k'_t) + \text{UCB}_t(k_t^*, k_t^{'*}) - \text{LCB}_t(k_t^*, k_t^{'*})$. Our choice of $(k_t, k'_t)$ ensures that $\min\{N^{k_t,s}, N^{k'_t,b}\} \leq \min\{N^{k_t^*,s}, N^{k_t^{'*},s}\}$ (Line 14). Then, by definition of UCB and LCB, and since the discretization is at most $O(\epsilon)$, we get $r(x_t, p_t) \lesssim \widetilde{O}\big(\sqrt{1/N^{k_t,s}} + \sqrt{1/N^{k'_t,b}} + \epsilon\big)$.

To bound the regret of the Successive Elimination phase, we consider separately the rounds (indexed by $\mathcal{T}_k^s$) where the bound is dominated by $\sqrt{1/N^{k,s}}$ and $k_t = k$, and the rounds (indexed by $\mathcal{T}_{k'}^b$) where it is dominated by $\sqrt{1/N^{k',b}}$ and $k'_t = k'$. Then, we decompose the regret of this phase as $\sum_{t \notin \mathcal{T}^{\text{par}} \cup \mathcal{T}^{\text{int}}} r(x_t, p_t) = \sum_{k \in \mathcal{K}} \big( \sum_{t \in \mathcal{T}_k^s} r(x_t, p_t) + \sum_{t \in \mathcal{T}_k^b} r(x_t, p_t) \big)$. Choosing $r_a = 2^{-a}$ and $r_{\overline{a}} \approx \epsilon$, we consider the decreasing sequence of intervals $(r_a)_{a \leq \overline{a}}$. The previous result implies that if $N^{k,s} \in [\widetilde{O}(1/r_a^2), \widetilde{O}(1/r_{a+1}^2)]$, then $r(x_t, p_t) \lesssim r_a$. Thus,

$$\sum_{t \in \mathcal{T}_k^s} r(x_t, p_t) \lesssim \widetilde{O}\bigg( r_1^2 + \sum_{1 \leq a \leq \overline{a}-1} r_a \left( r_{a+1}^2 - r_a^2 \right) + r_{\overline{a}} |\mathcal{T}_k^s| \bigg).$$

This sum is of order $\text{T}_k^s \epsilon + {\log(1/\delta)}/{\epsilon}$. We sum over $(k, k') \in \mathcal{K}$, and use the fact that $\sum_{k \in \mathcal{K}} \text{T}_k^s + \text{T}_k^b \leq T$ while $|\mathcal{K}| \approx \epsilon^{-1}$, to conclude that $R_T = \widetilde{O}(T^{2/3})$. $\qquad \square$

## 5    TRADING OFF BUDGET BALANCE CONSTRAINTS FOR FEEDBACK

In this section, we show that it is possible to relax the budget balance constraints in order to handle scenarios with limited feedback. In particular, we show that any EOC-like algorithm for two-bit feedback (for instance, Algorithms 1 and 3) can be suitably adapted to handle settings with one-bit feedback. To achieve this, we resort to the notion of *global budget balance* recently introduced by Bernasconi et al. (2024) and we show that budget can be exploited to compensate for the lack of feedback. The notion of global budget balance requires the learner to be budget balanced only "overall" (*i.e.*, over the whole time horizon). In particular, let $\text{PROFIT}_t(p, q) \coloneqq \mathbb{I}\{s_t \leq p\}\mathbb{I}\{q \leq b_t\}(q - p)$ be the profit extracted by the learner at time $t$ by posting prices $(p, q)$. Then, the learning algorithm is global budget balanced if the following inequality holds: $\sum_{t=1}^{T} \text{PROFIT}(p_t, q_t) \geq 0$.

Given an EOC-like algorithm for the two-bit strong budget balance set-up, the idea is to simulate its exploration rounds by doing separate updates for the seller and the buyer, each using a single bit of feedback. In particular, each time in which the two-bit algorithm uses the seller's feedback $\mathbb{I}\{s_t \leq p\}$, the one-bit algorithm needs to actively collect that information by posting $(p, -P)$ instead of $(p, p)$. Notice that, even under one-bit feedback, if the learner posts price $(p, -P)$ it is able to observe $\mathbb{I}\{s_t \leq p\}$ since the buyer is always going to accept the trade. Analogously, when the two-bit algorithm would employ $\mathbb{I}\{b_t \geq p\}$, the one-bit algorithm has to collect that information by

posting $(P, p)$. For instance, instead of executing EST-PAR, the one-bit algorithm runs two exploration phases: one in which it updates only $\widehat{\theta}^s$ (Line 4 of EST-PAR) by posting $(p_t, -P)$, and one in which it updates only $\widehat{\theta}^b$ by posting $(P, p_{t'})$. The same happens for the other estimation subroutines.

The one-bit algorithm simulates finer-grained feedback by posting pairs of prices $(P, p)$, $(p, -P)$, resulting in a negative GFT. By enforcing global budget balance, we allow the learning algorithm to set prices that are not budget balanced individually, as long as the overall budget balance is maintained. To do that, we add an initial "budget-collection" phase, allowing the algorithm to accumulate enough profit for subsequent exploration rounds. In the following, we demonstrate how sufficient profit can be accumulated during the budget-collection phase without incurring excessive regret relative to GFT. This allows us to provide sublinear regret guarantees under one-bit feedback, with rates depending on some natural characteristics of the underlying market. This approach is specifically suited for EOC-like algorithms, which require a limited number of learning rounds before starting to set greedy prices without requiring further feedback. By contrast, algorithms such as Algorithm 4 rely on two-bit feedback in each round to continuously refine their estimates of $D$ and $F$. Accumulating sufficient budget to emulate two-bit feedback throughout the entire time horizon would yield linear regret, rendering the reduction scheme inapplicable.

**Collecting budget.** Let $\mathcal{B}$ be the budget required by the one-bit algorithm to cover the rounds in which it must post either $(P, p)$ or $(p, -P)$ to collect information (the value of $\mathcal{B}$ will be set later). Therefore, the budget collection phase ends as soon as $\sum_t \text{PROFIT}(p_t, q_t) \geq \mathcal{B}$. We denote by $\tau$ the random variable indicating the last round of the budget-collection phase. Moreover, let $\mathcal{T}^{\text{B}} := [\![\tau]\!]$ be the random set of rounds designated for budget collection. For each round $t \in \mathcal{T}^{\text{B}}$, the algorithm samples a price $p_t$ uniformly from $[-P, P]$, and a value $i_t$ uniformly from $[0, \log T]$. Then, it posts a pair of prices $(p_t, q_t)$, where $q_t = p_t + 2^{-i_t}$. We start by providing a lower bound to the per-round expected profit obtained in this way.

**Lemma 2.** *For each round $t \in \mathcal{T}^{\text{B}}$ such that $b_t \geq s_t$, it holds:* $\mathbb{E}[\text{PROFIT}_t(p_t, q_t)] \geq \frac{(b_t - s_t)^2}{8P \log T} - \frac{2}{T}$, *where the expectation is with respect to the choice of $(p_t, q_t)$.*

Our guarantees for the one-bit feedback case will depend on instance-dependent properties of the market. We measure this by looking at the per-round GFT that be extracted on sufficiently long windows of time.

**Definition 1.** *Given a sequence of valuations $\{(s_t, b_t)\}_{t=1}^T$, $\alpha$ is defined as*

$$\alpha := \min_{t' \geq \log T} \sum_{t \in [\![t']\!]} \frac{[b_t - s_t]^+}{t'}.$$

Intuitively, $\alpha$ measures how easily the learner can accumulate budget. The budget-collection phase will have at least length of order $\log T$ since $\tau \geq \mathcal{B}/2P$, and $\mathcal{B} = \Omega(\log T)$ even in the setting without noise. Then, we can show that the profit accumulated by time $\tau$ is at least a factor of $1/\log T$ of the cumulative GFT up to that point, assuming all trades up to $\tau$ had occurred.

**Lemma 3.** *For $\tau \geq \log T$, it holds with probability at least $1 - 1/T$ that*

$$\sum_{t \in [\![\tau]\!]} \text{PROFIT}_t(p_t, q_t) \geq \frac{\alpha}{8P \log(T)} \sum_{t \in [\![\tau]\!]} [b_t - s_t]^+ - \sqrt{4P^2 \log(T) \sum_{t \in [\![\tau]\!]} [b_t - s_t]^+} - 2.$$

Now let $\mathcal{A}$ be a two-bit algorithm such that, with probability at least $1 - 1/T$, it requires at most $T^{\text{E}}$ exploration rounds, and has regret for the commit phase of $R_T^{(2\,\text{bit})}$. We set the overall budget to be collected to

$$\mathcal{B} = \max\{2048P^4\alpha^{-2}\log^3 T, 2PT^{\text{E}}\}.\,^3 \tag{5}$$

The first argument is used in the following proof to establish an upper bound on concentration terms, while the second accounts for the exploration costs over $T^{\text{E}}$ rounds. Then, the resulting one-bit learning algorithm has the following guarantees.

---

[3]With probability at most $1/T$, the budget $\mathcal{B}$ may be insufficient for $T^{\text{E}}$ exploration rounds. In this case, we halt exploration and begin posting arbitrary strong budget-balanced prices.

**Theorem 3.** *Given the two-bit algorithm $\mathcal{A}$, the corresponding one-bit learning algorithm satisfies global budget balance and, with probability at least $1 - 1/T$, has regret*

$$R_T^{(1\,\mathrm{bit})} \leq O\left(\alpha^{-3} T^{\mathrm{E}} \log^4 T\right) + R_T^{(2)}.$$

The guarantees above rely on the specific characteristics of the market, represented by the parameter $\alpha$. When $\alpha = \Omega(1)$ the market is well-behaved, meaning that trade opportunities arise frequently enough. This occurs, for example, when buyers are willing to pay more than the seller's asking price, even if only by a small margin. When $\alpha = \Omega(1)$, the guarantees of Theorem 3 can be effectively leveraged. For instance, under the set-up of Section 4, we can use Algorithm 3 as the two-bid algorithm $\mathcal{A}$. Then, $\mathcal{B} \leq 4PT^{\mathrm{E}} = 4P \cdot 2P(|\mathcal{T}_{T+1}^{\mathrm{par}}| + \mathrm{T}^{\mathrm{int}} + 2|\mathcal{K}|\mathrm{T}^{\mathrm{FD}})$, which is of order $\widetilde{O}(P^4 dT^{3/4})$. Therefore, the final regret bound in the noisy, one-bit setting with global budget balance is of order $\widetilde{O}(T^{3/4})$. A similar argument shows that in the noiseless setting of Section 3 the regret is of order $\mathrm{polylog}(T)$.

When the market is not well-behaved, we can still establish sublinear regret guarantees under one-bit feedback and global budget balance. Indeed, through steps analogous to those of Lemma 3, we can show that

$$\sum_{t \in \llbracket \tau \rrbracket} \mathrm{PROFIT}_t(p_t, q_t) \geq \Omega\left(\tau^{-1}\left(\sum_{t \in \llbracket \tau \rrbracket}([b_t - s_t]^+)\right)^2\right) - \widetilde{O}(\sqrt{T}).$$

Moreover, we can exploit the fact that we can decompose the one-bit regret as

$$R_T^{(1\,\mathrm{bit})} \leq \sum_{t \in \llbracket \tau \rrbracket}[b_t - s_t]^+ + \sqrt{16P^2\tau \log(T)} + 2T^{\mathrm{E}} + R_T^{(2)} \quad \text{(see proof of Theorem 3).}$$

Then, if we take the budget $\mathcal{B}$ (or, equivalently, $T^{\mathrm{E}}$) to be $\log T$, the expression above yields $\sum_{t \in \llbracket \tau \rrbracket}[b_t - s_t]^+ \leq \widetilde{O}(\tau^{1/2}T^{1/4})$. In the worst case we have $\tau = T$ (*i.e.,* the market does not allow to collect enough budget), which yields a regret of order $\widetilde{O}(T^{3/4})$ in the deterministic set-up of Section 3. By setting $\mathcal{B} \approx T^{3/4}$ we achieve a regret of $\widetilde{O}(T^{7/8})$ in the noisy set-up of Section 4. Clearly, the rates are worse compared to the case where $\alpha = \Omega(1)$, due to the difficulty in accumulating sufficient budget to cover the cost of posting non-budget balanced prices.

## ACKNOWLEDGMENTS

MB and MC are partially supported by the FAIR (Future Artificial Intelligence Research) project PE0000013, funded by the NextGenerationEU program within the PNRR-PE-AI scheme (M4C2, investment 1.3, line on Artificial Intelligence). MC is also partially supported by the EU Horizon project ELIAS (European Lighthouse of AI for Sustainability, No. 101120237). AC is partially supported by ERC grant (Project 101165466 — PLA-STEER). VP acknowledges support from the French National Research Agency (ANR) under grant number ANR-19-CE23- 0026 as well as from the grant "Investissements d'Avenir" (LabEx Ecodec/ANR-11-LABX-0047). SG gratefully acknowledges funding from the Jacques Hadamard Foundation (FMJH).

Funded by the European Union. Views and opinions expressed are however those of the author(s) only and do not necessarily reflect those of the European Union or the European Research Council Executive Agency. Neither the European Union nor the granting authority can be held responsible for them.

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

## A  BASIC FACTS ABOUT ELLIPSOIDS

An ellipsoid $E$ with center $c \in \mathbb{R}^d$ and positive definite matrix $\mathbf{M} \in \mathbb{R}^{d \times d}$ is defined as

$$E(\mathbf{M}, c) := \left\{ x \in \mathbb{R}^d : \|x - c\|_{\mathbf{M}^{-1}} \leq 1 \right\}.$$

Given an hyperplane $H(a, c) := \{x \in \mathbb{R}^d : a^\top(x - c) \leq 0\}$, the minimal volume ellipsoid containing $K = E(\mathbf{M}, c) \cap H(a, c)$ can be computed in closed form as follows. The new center is

$$c' = c - \frac{1}{d+1} \cdot \frac{1}{\sqrt{a^\top \mathbf{M} a}} \mathbf{M} a,$$

and the new positive definite matrix is

$$\mathbf{M}' = \frac{d^2}{d^2 - 1} \left( \mathbf{M} - \frac{2}{d+1} \cdot \frac{\mathbf{M} a a^\top \mathbf{M}}{a^\top \mathbf{M} a} \right).$$

Then, the Löwner-John ellipsoid $\mathrm{LJ}(K)$ of the half ellipsoid $K$ is the ellipsoid $E(\mathbf{M}', c')$.

Going from $E(\mathbf{M}, c)$ to $E(\mathbf{M}', c')$ the volume shrinks by the following amount:

$$\mathrm{vol}(E(\mathbf{M}', c')) \leq e^{-1/2d} \mathrm{vol}(E(\mathbf{M}, c)).$$

An in-depth analysis of these results can be found in Grötschel et al. (1993).

## B  PROOF OF PROPOSITION 1

Before proving Proposition 1, we begin by provide a high-level view of Algorithm 1. This algorithm maintains two ellipsoidal uncertainty sets $E_t^s$ and $E_t^b$ (see Appendix A for some basic facts about ellipsoids). The key idea is that, at each update, the volume of an uncertainty shrinks "fast enough" to yield a good estimate of the true parameter in a small number of iterations. The problems of computing the maximum and minimum possible valuations of the seller and buyer (Line 4) admit a simple closed-form solution (see, *e.g.*, , Grötschel et al. (1993, Chapter 3)).

At each round $t$, given the current uncertainty sets we compute the maximum and minimum possible valuations of the seller for the current product as

$$\underline{s_t} = \min_{\theta \in E_t^s} x_t^\top \theta \quad \text{and} \quad \overline{s_t} = \max_{\theta \in E_t^s} x_t^\top \theta. \tag{6}$$

The maximum and minimum possible valuations of the buyer, denoted by $\underline{b_t}$ and $\overline{b_t}$, are computed analogously. All of these optimization problems admit a simple closed form solution (see, *e.g.*, , Grötschel et al. (1993, Chapter 3)).

If the smallest possible valuation for the buyer $\underline{b_t}$ is above the highest possible valuation for the seller $\overline{s_t}$ (Line 5), the algorithm can post any price between these two values, ensuring that the trade will occur. If $\overline{s_t}$ and $\underline{s_t}$ are far apart (Line 6), the algorithm performs a binary search step (*a.k.a.* "explore" step), and posts a price $p_t$ which is halfway between the seller's minimum and the maximum possible valuations. Then, we update the seller's uncertainty set as follows: if the seller accepts the sale, it implies that $x_t^\top \theta^s \leq p_t$, and we define the half-ellipsoid $K_{t+1}^s = E_t^s \cap \{\theta : x_t^\top \theta^s \leq p_t\}$. Otherwise, we set $K_{t+1}^s = E_t^s \cap \{\theta : x_t^\top \theta^s > p_t\}$. Finally, we round this half-ellipsoid by replacing it by its Löwner-John ellipsoid, *i.e.*, the ellipsoid containing $K_{t+1}^s$ with the smallest volume. We proceed similarly for the buyer (Line 9). If the highest and lowest possible valuations of both the buyer and seller are close to each other, and the buyer's minimum valuation is lower than the seller's maximum valuation, then the difference of valuations between the two parties is small, resulting in a negligible gain from trade. Therefore, in this case, we can safely post an arbitrary price (Line 12).

**Proposition 1.** *Assume that the seller's and buyer's valuations follow Equation* (2). *Under Assumption 1, Algorithm 1 with $\epsilon = ABd^2/T$ has regret bounded by $R_T \leq 10ABd^2 \log\left(20(d+1)Td^{-2}\right)$.*

*Proof.* The proof of Proposition 1 relies on the fact that when the algorithm updates $E_t^s$ or $E_t^b$, it behaves as the ELLIPSOIDPRICING algorithm of Cohen et al. (2020). More precisely, we first note that under Assumption 1, $E_1^s$ and $E_1^b$ contain $\theta^s$ and $\theta^b$, respectively. Moreover, straightforward

induction shows that if $E_t^s$ contains $\theta^s$, then $K_{t+1}^s$ also contains $\theta^s$, and so does $E_{t+1}^s$ (similar reasoning applies to $\theta^b$ and $E_t^b$).

This implies that for all rounds, $\underline{s_t} \leq s_t \leq \overline{s_t}$, and $\underline{b_t} \leq b_t \leq \overline{b_t}$. In particular, if the condition in Line 5 is verified, we have $s_t \leq p_t \leq b_t$, and so the instantaneous regret at that round is 0.

We also underline that Algorithm 1 only explores to estimate $\theta^s$ (Line 6) if the condition

$$\max_{\theta \in E_t^s} x_t^\top \theta - \min_{\theta \in E_t^s} x_t^\top \theta \geq \epsilon$$

is verified, or equivalently if

$$\max_{\theta \in E_t^s} \frac{x_t^\top}{B} \theta - \min_{\theta \in E_t^s} \frac{x_t^\top}{B} \theta \geq \frac{\epsilon}{B}.$$

Since the normalized contexts $\frac{x_t^\top}{B}$ are bounded in norm by 1 by Assumption 1, we can apply Lemma 1 by Cohen et al. (2020). This Lemma states that Algorithm 1 will execute the steps in Lines 7-8 at most $2d^2 \log(20A(d+1)/\frac{\epsilon}{B})$ times, after which the condition in Line 6 will never be verified again. A similar reasoning allows us to bound the number of exploration rounds for the buyer's parameter $\theta^b$ (Line 9). Therefore, the total number of exploration rounds is bounded by $2 \times 2d^2 \log(20A(d+1)/\frac{\epsilon}{B})$. We notice that Assumption 1 also implies that the valuations are in $[-AB, AB]$, and so the instantaneous regret is bounded by $2AB$. Thus, the regret of the exploration phase (Lines 6 and 9) is bounded by $8ABd^2 \log(20A(d+1)/\frac{\epsilon}{B})$.

Finally, if the conditions on Lines 5, 6 and 9 are not verified, it implies that

$$\begin{aligned} b_t - s_t &\leq \overline{b_t} - \underline{s_t} \\ &\leq \underline{b_t} + \epsilon - \overline{s_t} + \epsilon \\ &\leq 2\epsilon. \end{aligned}$$

Then, the instantaneous regret at that round is at most $2\epsilon$, and the total regret for these steps is at most $2\epsilon T$. Choosing $\epsilon = \frac{ABd^2}{T}$ yields the result. $\square$

## C  PROOF OF THEOREM 1

In order to prove Theorem 1, we provide a more detailed version of the EXPLORE-OR-COMMIT algorithm in Algorithm 5. Note that we adopt the convention $1/0 = +\infty$.

Before analyzing the EXPLORE-OR-COMMIT algorithm, we define the various quantities used within it. Specifically, $\mathcal{T}_t^{\text{par}}$ represents the set of indices from previous rounds that were dedicated to estimating the parameters $\theta^s$ and $\theta^b$ (Line 8). We denote by $\widehat{\theta}_t^s$ and $\widehat{\theta}_t^b$ the estimates for the parameters, and $\mathbf{V}_t$ is the corresponding empirical covariance matrix at the beginning of round $t$. The set $\mathcal{T}^{\text{int}}$ contains the indices of the rounds used to estimate the integrals $I$ and $J$ (Line 13). Moreover, $\widehat{I}^k$ is the estimate of $I(k\epsilon)$, and $\widehat{J}^k$ is the estimate of $J(k\epsilon)$. The quantities $\widehat{F}^k$ and $\widehat{D}^k$ represent our estimates of $F^s(k\epsilon)$ and $D^b(k\epsilon)$, respectively.

We use the following lemma to bound the total duration of the parameter exploration phase (the proof can be found in Appendix E.2).

**Lemma 4.** *Almost surely, the length of exploration phase $\mathcal{T}_{T+1}^{\text{par}}$ is bounded as*

$$|\mathcal{T}_{T+1}^{\text{par}}| \leq \frac{d \log\left(\frac{T+d}{d}\right)}{\mu^2}.$$

The following lemma provides bounds on the estimation errors for $\theta^s$, $\theta^b$, $I$, $J$, $F^s$, and $D^b$ for the values of $\mu$, $\mathrm{T}^{\text{int}}$, and $\mathrm{T}^{\text{FD}}$ specified in Theorem 2. Let $\mathcal{E}$ be the event

$$\begin{aligned} \mathcal{E} := \Big\{ \forall k \in \mathcal{K}, \ \forall t \notin \mathcal{T}_T^{\text{par}} \cup \mathcal{T}^{\text{int}} \ \ &\left| x_t^\top (\widehat{\theta}_t^b - \theta^b) \right| \leq \epsilon \text{ and} \\ &\left| x_t^\top (\widehat{\theta}_t^s - \theta^s) \right| \leq \epsilon \text{ and} \\ &\left| \widehat{I}^k - I(k\epsilon) \right| \leq 2\epsilon \text{ and} \\ &\left| \widehat{J}^k - J(k\epsilon) \right| \leq 2\epsilon \Big\}. \end{aligned}$$

---

**Algorithm 5** EXPLORE-OR-COMMIT (detailed version)

---

1: **Input**: parameter $\mu > 0$, length of exploration phases $\mathrm{T}^{\mathrm{int}}$ and $\mathrm{T}^{\mathrm{FD}}$, discretization size $\epsilon$, confidence level $\delta$.

2: **Initialize**: $\mathcal{T}_1^{\mathrm{par}} = \mathcal{T}^{\mathrm{int}} = \varnothing$, $\widehat{\theta}_1^s = \mathbf{0}_d$, $\widehat{\theta}_1^b = \mathbf{0}_d$, $\mathbf{V}_1 = \mathbf{I}_d$, $K = \lceil 2P/\epsilon \rceil + 3$, $\mathcal{K} = [\![-K, K]\!]$, $\mathcal{T}_k^{\mathrm{F}} = \mathcal{T}_k^{\mathrm{D}} = \varnothing$ for all $k \in \mathcal{K}$, $\widehat{F}^k = \widehat{D}^k = 0$ for all $k \in \mathcal{K}$.

3: **while** $t \leq T$ **do**

4:     **if** $\|x_t\|_{\mathbf{V}_t^{-1}} > \mu$ **then**           ▷Estimate the parameters $\theta^b$ and $\theta^s$

5:         Draw $p_t \sim \mathcal{U}([-P, P])$ and post prices $(p_t, p_t)$

6:         Update $\mathcal{T}_{t+1}^{\mathrm{par}} \leftarrow \mathcal{T}_t^{\mathrm{par}} \cup \{t\}$

7:         Update $\mathbf{V}_{t+1} = \left( \sum_{l \in \mathcal{T}_{t+1}^{\mathrm{par}}} x_l x_l^\top + \mathbf{I}_d \right)^{-1}$

8:         Update parameter estimates

$$\widehat{\theta}_{t+1}^s = 2P\mathbf{V}_{t+1} \sum_{l \in \mathcal{T}_{t+1}^{\mathrm{par}}} \left( \mathbb{I}\{p_l \leq s_l\} - \frac{1}{2} \right) x_l, \quad \widehat{\theta}_{t+1}^b = 2P\mathbf{V}_{t+1} \sum_{l \in \mathcal{T}_{t+1}^{\mathrm{par}}} \left( \mathbb{I}\{p_l \leq b_l\} - \frac{1}{2} \right) x_l$$

9:     **else if** $|\mathcal{T}^{\mathrm{int}}| < \mathrm{T}^{\mathrm{int}}$ **then**         ▷Estimate the integrals $I$ and $J$

10:         Draw $p_t \sim \mathcal{U}([-P, P])$ and post prices $(p_t, p_t)$

11:         Update $\mathcal{T}^{\mathrm{int}} \leftarrow \mathcal{T}^{\mathrm{int}} \cup \{t\}$

12:         **if** $|\mathcal{T}^{\mathrm{int}}| = \mathrm{T}^{\mathrm{int}}$ **then**

13:             Compute estimate of integrals $I$ and $J$

14:             **for** $k \in \mathcal{K}$ **do**

$$\widehat{I}^k = \frac{2P}{\mathrm{T}^{\mathrm{int}}} \sum_{l \in \mathcal{T}^{\mathrm{int}}} \mathbb{I}\left\{ k\epsilon + x_l^\top \widehat{\theta}^b \leq p_l \leq b_l \right\}, \quad \widehat{J}^k = \frac{2P}{\mathrm{T}^{\mathrm{int}}} \sum_{l \in \mathcal{T}^{\mathrm{int}}} \mathbb{I}\left\{ s_l \leq p_l \leq k\epsilon + x_l^\top \widehat{\theta}^s \right\}.$$

15:     **else if** for some $k \in \mathcal{K}$, $|\mathcal{T}_k^{\mathrm{F}}| < \mathrm{T}^{\mathrm{FD}}$ **then**         ▷Estimate $F_k$

16:         Set $p_t = x_t^\top \widehat{\theta}^s + k\epsilon$ and post $(p_t, p_t)$

17:         Update $\mathcal{T}_k^{\mathrm{F}} \leftarrow \mathcal{T}_k^{\mathrm{F}} \cup \{t\}$

18:         **if** $|\mathcal{T}_k^{\mathrm{F}}| = \mathrm{T}^{\mathrm{FD}}$ **then** set

$$\widehat{F}^k = \tfrac{2P}{\mathrm{T}^{\mathrm{FD}}} \textstyle\sum_{l \in \mathcal{T}_k^{\mathrm{F}}} \mathbb{I}\left\{ s_l \leq p_l \right\}$$

19:     **else if** for some $k \in \mathcal{K}$, $|\mathcal{T}_k^{\mathrm{D}}| < \mathrm{T}^{\mathrm{FD}}$ **then**         ▷Estimate $F_k$

20:         Set $p_t = x_t^\top \widehat{\theta}^b + k\epsilon$ and post $(p_t, p_t)$

21:         Update $\mathcal{T}_k^{\mathrm{D}} \leftarrow \mathcal{T}_k^{\mathrm{D}} \cup \{t\}$

22:         **if** $|\mathcal{T}_k^{\mathrm{D}}| = \mathrm{T}^{\mathrm{FD}}$ **then** set

$$\widehat{D}^k = \tfrac{2P}{\mathrm{T}^{\mathrm{FD}}} \textstyle\sum_{l \in \mathcal{T}_k^{\mathrm{D}}} \mathbb{I}\left\{ p_l \leq b_l \right\}$$

23:     **else**                                  ▷Post greedy price

24:         set $\mathcal{A}_t = \left\{ (k, k') \in \mathcal{K}^2 : k' = \left\lfloor \frac{x_t^\top \left( \widehat{\theta}_t^s - \widehat{\theta}_t^b \right) + k\epsilon}{\epsilon} \right\rfloor \right\}$

25:         $k_t = \arg\max_{k : \exists k', (k,k') \in \mathcal{A}_t} \widehat{I}^{k'} \widehat{F}^k + \widehat{J}^k \widehat{D}^{k'}$.

26:         Set $p_t = x_t^\top \widehat{\theta}_t^s + k_t \epsilon$ and post $(p_t, p_t)$

---

Let us define the exploitation (*i.e.,* commit) phase as

$$\mathcal{T}^{\text{C}} := \llbracket T \rrbracket \setminus \left( \mathcal{T}_T^{\text{par}} \cup \mathcal{T}^{\text{int}} \bigcup_{k \in \mathcal{K}} \left( \mathcal{T}_k^{\text{F}} \cup \mathcal{T}_k^{\text{D}} \right) \right).$$

Moreover, let $\mathcal{E}^{\text{EOC}}$ be the event

$$\mathcal{E}^{\text{EOC}} := \mathcal{E} \cap \left\{ \forall k \in \mathcal{K}, \ \forall t \in \mathcal{T}^{\text{C}}, \left| \widehat{F}^k - F^s(k\epsilon) \right| \le (L+1)\epsilon \text{ and } \left| \widehat{D}^k - D^s(k\epsilon) \right| \le (L+1)\epsilon \right\}.$$

Then, we can lower bound the probability of event $\mathcal{E}^{\text{EOC}}$ as follows (the proof is postponed to Appendix E.3).

**Lemma 5.** *For the choice* $\mu = \epsilon \left( P\sqrt{d \log \left( \frac{1 + B^2 T}{\delta} \right)} + A \right)^{-1}$, $\text{T}^{\text{int}} = 8P^2 \log(1/\delta)/\epsilon^2$, *and* $\text{T}^{\text{FD}} = 2 \log(1/\delta)\epsilon^{-2}$, *it holds that*

$$\mathbb{P} \left( \mathcal{E}^{\text{EOC}} \right) \ge 1 - 2\delta - 8\delta(2K+1).$$

Note that the choice $\delta = \left( T(74 + 32P\epsilon^{-1}) \right)^{-1}$ ensures that the event $\mathcal{E}^{\text{EOC}}$ happens with probability at least $1 - T^{-1}$.

The following Lemma bounds the error for estimating the gain from trade of a price $p = x_t^\top \widehat{\theta}_t^s + k\epsilon$ on the high-probability event $\mathcal{E}^{\text{EOC}}$ (the proof is presented in Appendix E.4).

**Lemma 6.** *On the event* $\mathcal{E}^{\text{EOC}}$, *for all* $t \in \mathcal{T}^{\text{C}}$, *and all* $(k, k') \in \mathcal{A}_t$,

$$\left| \widehat{I}^{k'} \widehat{F}^k + \widehat{J}^k \widehat{D}^{k'} - \text{EGFT}(x_t, x_t^\top \widehat{\theta}_t^s + k\epsilon) \right| \le (10PL + 4P + 7)\epsilon.$$

Next, we bound the discretization error. Let us define

$$(k_t^*, k_t'^*) \in \arg\max_{(k,k') \in \mathcal{A}_t} \text{EGFT}(x_t, x_t^\top \widehat{\theta}_t^s + k\epsilon).$$

Then, the following result holds (see Appendix E.5 for its proof).

**Lemma 7.** *On the event* $\mathcal{E}^{\text{EOC}}$, *we have that*

$$\left| \max_p \text{EGFT}(x_t, p) - \text{EGFT}(x_t, x_t^\top \widehat{\theta}_t^s + k_t^*\epsilon) \right| \le 2LP\epsilon.$$

We are now ready to prove Theorem 1. For $p \in \mathbb{R}$, $x \in \mathbb{R}^d$, we define

$$\Delta(x, p) := \max_{p'} \text{EGFT}(x, p') - \text{EGFT}(x, p).$$

We begin by decomposing the regret as

$$R_T = \sum_{t \in \mathcal{T}_{T+1}^{\text{par}}} \Delta(x_t, p_t) + \sum_{t \in \mathcal{T}^{\text{int}}} \Delta(x_t, p_t) +$$

$$\sum_{k \in \mathcal{K}} \left( \sum_{t \in \mathcal{T}_k^{\text{F}}} \Delta(x_t, p_t) + \sum_{t \in \mathcal{T}_k^{\text{D}}} \Delta(x_t, p_t) \right) + \sum_{t \in \mathcal{T}^{\text{C}}} \Delta(x_t, p_t).$$

Using the fact that $\Delta(x_t, p_t) \le 2P$, we obtain

$$R_T \le 2P \left( |\mathcal{T}_{T+1}^{\text{par}}| + \text{T}^{\text{int}} + 2|\mathcal{K}|\text{T}^{\text{FD}} \right) + \sum_{t \in \mathcal{T}^{\text{C}}} \Delta(x_t, p_t).$$

On the one hand, Lemma 4 implies that $|\mathcal{T}_{T+1}^{\text{par}}| \le \frac{d \log \left( \frac{T+d}{d} \right)}{\mu^2}$. For the choice $\mu = \epsilon \left( P \left( d \log \left( \frac{1+B^2 T}{\delta} \right) \right)^{1/2} + A \right)^{-1}$, $\text{T}^{\text{int}} = 8P^2 \log(1/\delta)\epsilon^{-2}$, and $\text{T}^{\text{FD}} = 2 \log(1/\delta)\epsilon^{-2}$, this implies that

$$R_T \le 2Pd \log \left( \frac{T+d}{d} \right) \left( P\sqrt{d \log \left( \frac{1+B^2 T}{\delta} \right)} + A \right)^2 \epsilon^{-2} + 16P^3 \log(1/\delta)\epsilon^{-2} +$$

$$8P \left( \frac{4P}{\epsilon} + 9 \right) \log(1/\delta)\epsilon^{-2} + \sum_{t \in \mathcal{T}^{\text{C}}} \Delta(x_t, p_t).$$

Note that the first term is of order $\epsilon^{-3}$ when $\epsilon$ is small enough. On the other hand, on the event $\mathcal{E}^{\text{EOC}}$, by Lemma 7 we have

$$
\begin{aligned}
\sum_{t \in \mathcal{T}^{\text{c}}} & \Delta(x_t, p_t) \\
&= \sum_{t \in \mathcal{T}^{\text{c}}} \Delta(x_t, x_t^\top \widehat{\theta}_t^s + k_t^* \epsilon) + \sum_{t \in \mathcal{T}^{\text{c}}} \text{EGFT}(x_t, x_t^\top \widehat{\theta}_t^s + k_t^* \epsilon) - \text{EGFT}(x_t, x_t^\top \widehat{\theta}_t^s + k_t \epsilon) \\
&\leq 2TLP\epsilon + \sum_{t \in \mathcal{T}^{\text{c}}} \Big( \text{EGFT}(x_t, x_t^\top \widehat{\theta}_t^s + k_t^* \epsilon) - \text{EGFT}(x_t, x_t^\top \widehat{\theta}_t^s + k_t \epsilon) \Big),
\end{aligned}
$$

Moreover, our choice of $k_t$ ensures that for $k_t'$ such that $(k_t, k_t') \in \mathcal{A}_t$,

$$
\widehat{I}^{k_t'} \widehat{F}^{k_t} + \widehat{J}^{k_t} \widehat{D}^{k_t'} \geq \widehat{I}^{k_t'^*} \widehat{F}^{k_t^*} + \widehat{J}^{k_t^*} \widehat{D}^{k_t'^*}.
$$

By Lemma 6, this implies that, on event $\mathcal{E}^{\text{EOC}}$,

$$
\text{EGFT}(x_t, x_t^\top \widehat{\theta}_t^s + k_t^* \epsilon) - \text{EGFT}(x_t, x_t^\top \widehat{\theta}_t^s + k_t \epsilon) \leq (20PL + 8P + 14)\epsilon.
$$

Thus, on the event $\mathcal{E}^{\text{EOC}}$,

$$
\begin{aligned}
R_T \leq & 2Pd \log\left(\frac{T+d}{d}\right) \left(P\sqrt{d\log\left(\frac{1+B^2T}{\delta}\right)} + A\right)^2 \epsilon^{-2} + 16P^3 \log(1/\delta)/\epsilon^2 \\
& + 8P\left(\frac{P}{\epsilon} + 3\right) \log(1/\delta)\epsilon^{-2} + 2TLP\epsilon + T(20PL + 8P + 14)\epsilon.
\end{aligned}
$$

Substituting the values of $\epsilon$ and $\delta$ as per Theorem 1 and using Lemma 5 allows us to conclude the proof. In particular, the result in Theorem 1 holds with $\tilde{C} = 22PL + 26P + 15$, provided that $T \geq 74 + 32P$, that $T \geq 1 + B^2$, that $T \geq d$, $5P^2 d \log(T) \geq 1$, and that $160P^2 d^2 + 36P^3 \leq T^{1/4} \log(T)^{5/4}$.

## D  PROOF OF THEOREM 2

### D.1  DETAILED ALGORITHM

In order to prove Theorem 2, we provide a more detailed version of the SCOUTING BANDIT WITH INFORMATION POOLING algorithm in Algorithm 6. Note that in Lines 16 an 17, we adopt the convention $1/0 = +\infty$ when computing the upper- and lower- confidence bounds for increment levels $k, k'$ that have not yet been selected.

Before analyzing the SBIP algorithm, we introduce the different quantities appearing in the algorithm. In words, $\mathcal{T}_t^{\text{par}}$ is the set of indices of the previous rounds that have been spent estimating the parameters $\theta^s$ and $\theta^b$ at the beginning of round $t$. Similarly, $\widehat{\theta}_t^s$, $\widehat{\theta}_t^b$, and $\mathbf{V}_t$ are the estimates of the parameters and the corresponding empirical covariance matrix at the beginning of round $t$. The set $\mathcal{T}^{\text{int}}$ consists of the indices of the rounds used to estimate the integrals $I$ and $J$. Moreover, $\widehat{I}^k$ estimates $I(k\epsilon)$, and $\widehat{J}^k$ estimates $J(k\epsilon)$. The quantities $\widehat{F}_t^k$ and $\widehat{D}_t^k$ represent our estimates of $F^s(k\epsilon)$ and $D^b(k\epsilon)$ at the beginning of round $t$. Similarly, $N_t^{s,k}$ (resp., $N_t^{b,k}$) counts the number of rounds in the successive elimination phase where the increment $p_l - x_l^\top \widehat{\theta}_l^s$ was equal to $k\epsilon$ (resp., where the increment $p_l - x_l^\top \widehat{\theta}_l^b$ was close to $k\epsilon$), and where we gained information on $F^s(k\epsilon)$ (resp., on $D^b(k\epsilon)$). The set $\mathcal{A}_t$ collects the pairs $(k, k') \in \mathcal{K}$ such that $x_t^\top \widehat{\theta}_t^s + k\epsilon \approx x_t^\top \widehat{\theta}_t^b + k'\epsilon$ is a possible price (within $[-P, P]$). Then, for $(k, k') \in \mathcal{A}_t$, the quantities $\text{UCB}_t(k, k')$ and $\text{LCB}_t(k, k')$ provide upper and lower confidence bounds on the expected gain from trade corresponding to this price.

The third phase is a *successive elimination phase*: at each round, we consider a set of possible optimal prices, with corresponding upper confidence bound larger than the highest lower confidence bound. This set is denoted by $\mathcal{K}_t$. In order to ensure sufficient exploration of all potentially optimal price increments $(k\epsilon, k'\epsilon)$ for $(k, k') \in \mathcal{K}_t$, we choose the price increment with the widest confidence interval: this corresponds to choosing the pair $(k, k') \in \mathcal{K}_t$ such that either $N_t^{s,k}$ or $N_t^{b,k'}$

---

**Algorithm 6** SCOUTING BANDITS WITH INFORMATION POOLING (SBIP) (detailed version)

---

1: **Input**: parameter $\mu > 0$, length of scouting phase $T^{\text{int}}$, discretization size $\epsilon$, confidence level $\delta$.

2: **Initialize**: $\mathcal{T}_1^{\text{par}} = \mathcal{T}^{\text{int}} = \varnothing$, $\widehat{\theta}_1^s = \mathbf{0}_d$, $\widehat{\theta}_1^b = \mathbf{0}_d$, $\mathbf{V}_1 = \mathbf{I}_d$, $K = \lceil 2P/\epsilon \rceil + 3$, $\mathcal{K} = [\![-K, K]\!]$,
   $\mathcal{T}_{s,k}^{\text{SE}} = \mathcal{T}_{b,k}^{\text{SE}} = \varnothing$ for all $k \in \mathcal{K}$, $N_1^{s,k} = N_1^{b,k} = 0$ for all $k \in \mathcal{K}$, $\widehat{F}_1^k = \widehat{D}_1^k = 0$ for all $k \in \mathcal{K}$.

3: **while** $t \leq T$ **do**

4:      **if** $\|x_t\|_{\mathbf{V}_t^{-1}} > \mu$ **then**                          ▷Estimate the parameters $\theta^b$ and $\theta^s$

5:          Draw $p_t \sim \mathcal{U}([-P, P])$ and post $(p_t, p_t)$

6:          Update $\mathcal{T}_{t+1}^{\text{par}} = \mathcal{T}_t^{\text{par}} \cup \{t\}$

7:          Update $\mathbf{V}_{t+1} = \left( \sum_{l \in \mathcal{T}_{t+1}^{\text{par}}} x_l x_l^\top + \mathbf{I}_d \right)^{-1}$

8:          Update parameter estimates

$$\widehat{\theta}_{t+1}^s = 2P\mathbf{V}_{t+1} \sum_{l \in \mathcal{T}_{t+1}^{\text{par}}} \left( \mathbb{I}\{p_l \leq s_l\} - \tfrac{1}{2} \right) x_l, \quad \widehat{\theta}_{t+1}^b = 2P\mathbf{V}_{t+1} \sum_{l \in \mathcal{T}_{t+1}^{\text{par}}} \left( \mathbb{I}\{p_l \leq b_l\} - \tfrac{1}{2} \right) x_l$$

9:      **else if** $|\mathcal{T}^{\text{int}}| < T^{\text{int}}$ **then**                       ▷Estimate the integrals $I$ and $J$

10:          Draw $p_t \sim \mathcal{U}([-P, P])$ and post $(p_t, p_t)$

11:          Update $\mathcal{T}^{\text{int}} = \mathcal{T}^{\text{int}} \cup \{t\}$

12:          **if** $|\mathcal{T}^{\text{int}}| = T^{\text{int}}$ **then**

13:              **for** $k \in \mathcal{K}$ **do**

$$\widehat{I}^k = \tfrac{2P}{T^{\text{int}}} \sum_{l \in \mathcal{T}^{\text{int}}} \mathbb{I} \left\{ k\epsilon + x_l^\top \widehat{\theta}^b \leq p_l \leq b_l \right\}, \quad \widehat{J}^k = \tfrac{2P}{T^{\text{int}}} \sum_{l \in \mathcal{T}^{\text{int}}} \mathbb{I} \left\{ s_l \leq p_l \leq k\epsilon + x_l^\top \widehat{\theta}^s \right\}$$

14:      **else**                                             ▷Run Successive Elimination

15:          Set $\mathcal{A}_t = \left\{ (k, k') \in \mathcal{K}^2 : k' = \left\lfloor \frac{(x_t^\top (\widehat{\theta}_t^s - \widehat{\theta}_t^b) + k\epsilon)}{\epsilon} \right\rfloor \right\}$

16:          **for** $(k, k') \in \mathcal{A}_t$ **do**

$$\text{UCB}_t(k, k') = \widehat{I}^{k'} \widehat{F}_t^k + \widehat{J}^k \widehat{D}_t^{k'} + (12PL + 7)\epsilon + 2P \left( \sqrt{2\log(1/\delta)/N_t^{s,k}} + \sqrt{2\log(1/\delta)/N_t^{b,k'}} \right),$$

$$\text{LCB}_t(k, k') = \widehat{I}^{k'} \widehat{F}_t^k + \widehat{J}^k \widehat{D}_t^{k'} - (12PL + 7)\epsilon - 2P \left( \sqrt{2\log(1/\delta)/N_t^{s,k}} + \sqrt{2\log(1/\delta)/N_t^{b,k'}} \right)$$

17:          Set $\mathcal{K}_t = \{(k, k') \in \mathcal{A}_t : \text{UCB}_t(k, k') \geq \max\{\text{LCB}_t(l, l') \ : \ (l, l') \in \mathcal{A}_t\}\}$

18:          Set $k_t^s = \arg\min \left\{ N_t^{s,k} : (k, k') \in \mathcal{K}_t \right\}$, and $k_t^b = \arg\min \left\{ N_t^{b,k'} : (k, k') \in \mathcal{K}_t \right\}$

19:          **if** $N_t^{s,k_t^s} \leq N_t^{b,k_t^b}$ **then**

20:              $k_t = k_t^s$, and set $k_t'$ to be such that $(k_t, k_t') \in \mathcal{K}_t$

21:              $\mathcal{T}_{s,k_t}^{\text{SE}} = \mathcal{T}_{s,k_t}^{\text{SE}} \cup \{t\}$

22:          **else**

23:              $k_t' = k_t^b$, and set $k_t$ to be such that $(k_t, k_t') \in \mathcal{K}_t$

24:              $\mathcal{T}_{b,k_t'}^{\text{SE}} = \mathcal{T}_{b,k_t'}^{\text{SE}} \cup \{t\}$

25:          Set $p_t = x_t^\top \widehat{\theta}_t^s + k_t\epsilon$ and post $(p_t, p_t)$

26:          Update $\widehat{F}_{t+1}^{k_t} = \frac{N_t^{k_t,s} \widehat{F}_t^{k_t} + \mathbb{I}\{s_t \leq p_t\}}{N_t^{k_t,s} + 1}$, $\widehat{D}_{t+1}^{k_t'} = \frac{N_t^{k_t',b} \widehat{D}_t^{k_t'} + \mathbb{I}\{b_t \geq p_t\}}{N_t^{k_t',b} + 1}$

27:          Update $N_{t+1}^{k_t,s} = N_t^{k_t,s} + 1$, and $N_{t+1}^{k_t',b} = N_t^{k_t',b} + 1$

28:      Quantities that have not been updated during round $t$ are kept the same for round $t + 1$

---

is the lowest of the set. We denote $(k_t, k'_t)$ the pair of increments chosen in such way. In order to analyze this phase, we store the indexes of the rounds where we chose this pair because $N_t^{s,k_t}$ was the smallest in the set $\mathcal{T}_{s,k_t}^{\text{SE}}$. Analogously, the rounds where we chose this pair because $N_t^{b,k'_t}$ was the smallest are stored in the set $\mathcal{T}_{b,k'_t}^{\text{SE}}$).

## D.2  REGRET ANALYSIS

The beginning of the proof of Theorem 2 is similar to that of Theorem 1. As in the previous case, we define the event

$$\mathcal{E} := \left\{ \forall k \in \mathcal{K}, \forall t \notin \mathcal{T}_{T+1}^{\text{par}} \cup \mathcal{T}^{\text{int}}, \quad \left| x_t^\top (\widehat{\theta}_t^b - \theta^b) \right| \leq \epsilon, \left| x_t^\top (\widehat{\theta}_t^s - \theta^s) \right| \leq \epsilon, \right.$$
$$\left. \left| \widehat{I}^k - I(k\epsilon) \right| \leq 2\epsilon, \left| \widehat{J}^k - J(k\epsilon) \right| \leq 2\epsilon \right\}.$$

Moreover, we define a new event $\mathcal{E}^{\text{SBIP}}$ as

$$\mathcal{E}^{\text{SBIP}} := \mathcal{E} \bigcap \left\{ \forall k \in \mathcal{K}, \forall t \notin \mathcal{T}_{T+1}^{\text{par}} \cup \mathcal{T}^{\text{int}}, \quad \left| \widehat{F}_t^k - F^s(k\epsilon) \right| \leq \sqrt{\frac{2\log(1/\delta)}{N_t^{s,k}}} + L\epsilon, \right.$$
$$\left. \left| \widehat{D}^k - D^s(k\epsilon) \right| \leq \sqrt{\frac{2\log(1/\delta)}{N_t^{b,k}}} + 2L\epsilon \right\}.$$

The following Lemma shows that the event $\mathcal{E}^{\text{SBIP}}$ happens with large probability. The detailed proof can be found in Appendix E.6.

**Lemma 8.** *For the choice* $\mu = \epsilon \left( P \left( d\log \left( \frac{1+B^2 T}{\delta} \right) \right)^{1/2} + A \right)^{-1}$ *and* $\mathrm{T}^{\text{int}} = 8P^2 \log(1/\delta)/\epsilon^2$, *it holds that*

$$\mathbb{P}\left( \mathcal{E}^{\text{SBIP}} \right) \geq 1 - 2\delta - 4\delta(2K+1) - 4\delta(2K+1)T.$$

Note that the choice $\delta = \left( T(38 + 16P\epsilon^{-1} + 16P\epsilon^{-1}T + 36T) \right)^{-1}$ ensures that the event $\mathcal{E}^{\text{SBIP}}$ happens with probability at least $1 - T^{-1}$.

The following Lemma (whose proof is located in Appendix E.7) shows that the upper and lower confidence bounds used in Algorithm 6 hold conditioned on the high-probability event $\mathcal{E}^{\text{SBIP}}$.

**Lemma 9.** *Under the assumptions of Lemma 8 and conditioned on the event* $\mathcal{E}^{\text{SBIP}}$, *we have that for all* $t \notin \mathcal{T}_{T+1}^{\text{par}} \cup \mathcal{T}^{\text{int}}$, *and all* $(k, k') \in \mathcal{A}_t$:

$$\text{LCB}_t(k, k') \leq \text{EGFT}(x_t, x_t^\top \widehat{\theta}_t^s + k\epsilon) \leq \text{UCB}_t(k, k').$$

*Moreover, it holds that* $(k_t^*, k_t'^*) \in \mathcal{K}_t$, *where we recall that*

$$(k_t^*, k_t'^*) \in \arg\max_{(k,k') \in \mathcal{A}_t} \text{EGFT}(x_t, x_t^\top \widehat{\theta}_t^s + k\epsilon).$$

Finally, we bound the number of times a sub-optimal price increment $k\epsilon$ can be selected. For $p \in \mathbb{R}$, $x \in \mathbb{R}^d$, recall that we defined

$$r(x, p) = \max_{p'} \text{EGFT}(x, p') - \text{EGFT}(x, p).$$

**Lemma 10.** *Conditioned on the event* $\mathcal{E}^{\text{SBIP}}$, *if* $t \in \mathcal{T}_{s,k}^{\text{SE}}$, *then the following condition holds*

$$r(x_t, p_t) \leq (50PL + 28)\epsilon + 16P\sqrt{\frac{2\log(1/\delta)}{N_t^{s,k}}}.$$

*Similarly, if* $t \in \mathcal{T}_{b,k'}^{\text{SE}}$, *then the following condition holds*

$$r(x_t, p_t) \leq (50PL + 28)\epsilon + 16P\sqrt{\frac{2\log(1/\delta)}{N_t^{b,k'}}}.$$

The proof for this result can be found in Appendix E.8.

We are now ready to bound the regret of SBIP. We begin by decomposing the regret as

$$R_T = \sum_{t \in \mathcal{T}^{\text{par}}_{T+1}} r(x_t, p_t) + \sum_{t \in \mathcal{T}^{\text{int}}} r(x_t, p_t) + \sum_{t \notin \mathcal{T}^{\text{par}}_{T+1} \cup \mathcal{T}^{\text{int}}} r(x_t, p_t).$$

Since the gain from trade is bounded by $2P$, this implies

$$R_T \le 2P|\mathcal{T}^{\text{par}}_{T+1}| + 2P|\mathcal{T}^{\text{int}}| + \sum_{t \notin \mathcal{T}^{\text{par}}_{T+1} \cup \mathcal{T}^{\text{int}}} r(x_t, p_t).$$

Then, using Lemma 4, by setting $\mu = \epsilon \left( P\left(d \log\left(\frac{1+B^2 T}{\delta}\right)\right)^{1/2} + A \right)^{-1}$ and $\mathrm{T}^{\text{int}} = 8P^2 \log(1/\delta)/\epsilon^2$ we have

$$R_T \le \epsilon^{-2} \left( 2Pd \log\left(\frac{T+d}{d}\right) \left( P\sqrt{d \log\left(\frac{1+B^2 T}{\delta}\right)} + A \right)^2 \right) +$$
$$16\epsilon^{-2} P^3 \log(1/\delta) + \sum_{t \notin \mathcal{T}^{\text{par}}_{T+1} \cup \mathcal{T}^{\text{int}}} r(x_t, p_t). \quad (7)$$

To bound the third term (*i.e.,* the regret of the successive elimination phase), we decompose it as

$$\sum_{t \notin \mathcal{T}^{\text{par}}_{T+1} \cup \mathcal{T}^{\text{int}}} r(x_t, p_t \epsilon) = \sum_{k \in \mathcal{K}} \sum_{t \in \mathcal{T}^{\text{SE}}_{s,k}} r(x_t, p_t) + \sum_{k' \in \mathcal{K}} \sum_{t \in \mathcal{T}^{\text{SE}}_{b,k'}} r(x_t, p_t).$$

For $k \in \mathcal{K}$, let us partition $\mathcal{T}^{\text{SE}}_{s,k}$ as follows. We denote by $T^{\text{SE}}_{s,k} = |\mathcal{T}^{\text{SE}}_{s,k}|$, and we define $t_1, \ldots, t_{T^{\text{SE}}_{s,k}}$ the rounds where $t \in \mathcal{T}^{\text{SE}}_{s,k}$. More formally, we have $t_1 < t_2 < \ldots < t_{T^{\text{SE}}_{s,k}}$, and $\{t_1, \ldots, t_{T^{\text{SE}}_{s,k}}\} = \mathcal{T}^{\text{SE}}_{s,k}$. We define $\overline{a} = \lfloor -\log_2 (2(50PL + 28)\epsilon) \rfloor$, and for $a \in [\![1, \overline{a}]\!]$, we define

$$\mathsf{t}_a = 2 \log(1/\delta) \cdot (32P2^a)^2 .$$

With these notation, we have

$$\sum_{t \in \mathcal{T}^{\text{SE}}_{s,k}} r(x_t, p_t) = \sum_{i \le \mathsf{t}_1 \wedge T^{\text{SE}}_{s,k}} r(x_{t_i}, p_{t_i})$$
$$+ \sum_{a=1}^{\overline{a}-1} \sum_{\mathsf{t}_a \wedge T^{\text{SE}}_{s,k} < i \le \mathsf{t}_{a+1} \wedge T^{\text{SE}}_{s,k}} r(x_{t_i}, p_{t_i}) + \sum_{(\mathsf{t}_{\overline{a}}+1) \wedge T^{\text{SE}}_{s,k} \le i \le T^{\text{SE}}_{s,k}} r(x_{t_i}, p_{t_i}).$$

For $1 \le a \le \overline{a}$, if $\mathsf{t}_a \le i \le T^{\text{SE}}_{s,k}$, by definition of $\mathsf{t}_a$, we have

$$(50PL + 28)\epsilon \le 16P \sqrt{\frac{2 \log(1/\delta)}{N^{s,k}_{t_i}}} \le \frac{2^{-a}}{2},$$

and Lemma 10 implies that, conditioned on the high-probability event $\mathcal{E}^{\text{SBIP}}$,

$$r(x_{t_i}, p_{t_i}) \le 2^{-a}.$$

Using the fact that $r(x_t, p_t) \le 2P$, we obtain

$$\sum_{t \in \mathcal{T}^{\text{SE}}_{s,k}} r(x_t, p_t) \le 2P\mathsf{t}_1 + \sum_{a=1}^{\overline{a}-1} 2^{-a} \left(\mathsf{t}_{a+1} - \lceil \mathsf{t}_a \rceil + 1\right) + 2^{-\overline{a}} \cdot T^{\text{SE}}_{s,k}$$

$$\le 2P\mathsf{t}_1 + \sum_{a=1}^{\overline{a}-1} 2^{-a} \left(\mathsf{t}_{a+1} - \mathsf{t}_a + 1\right) + 2^{-\overline{a}} \cdot T^{\text{SE}}_{s,k}.$$

Then,

$$
\begin{aligned}
\sum_{a=1}^{\overline{a}-1} 2^{-a} \left(\mathfrak{t}_{a+1} - \mathfrak{t}_a\right) &= 6\log(1/\delta) \cdot (32P)^2 \sum_{a=1}^{\overline{a}-1} 2^a \\
&= 6\log(1/\delta) \cdot (32P)^2 \, 2^{\overline{a}} \\
&\leq \frac{3\log(1/\delta) \cdot (32P)^2}{(50PL+28)\epsilon}
\end{aligned}
$$

Moreover, we have that

$$
2^{-\overline{a}} \cdot T_{s,k}^{\mathrm{SE}} \leq 2(50PL+28)\epsilon T_{s,k}^{\mathrm{SE}}.
$$

Therefore,

$$
\sum_{t \in \mathcal{T}_{s,k}^{\mathrm{SE}}} r(x_t, p_t) \leq 4P\log(1/\delta) \cdot (128P)^2 + \frac{3\log(1/\delta) \cdot (32P)^2}{(50PL+28)\epsilon} + 1 + 2(50PL+28)\epsilon T_{s,k}^{\mathrm{SE}}.
$$

We proceed similarly to bound $\sum_{t \in \mathcal{T}_{b,k}^{\mathrm{SE}}} r(x_t, p_t)$, and we obtain that

$$
\sum_{t \in \mathcal{T}_{b,k}^{\mathrm{SE}}} r(x_t, p_t) \leq 4P\log(1/\delta) \cdot (128P)^2 + \frac{3\log(1/\delta) \cdot (32P)^2}{(50PL+28)\epsilon} + 1 + 2(50PL+28)\epsilon T_{b,k}^{\mathrm{SE}}.
$$

Summing over $k \in \mathcal{K}$ and $k' \in \mathcal{K}$, and using the fact that $\sum_{k \in \mathcal{K}} T_{s,k}^{\mathrm{SE}} + T_{b,k}^{\mathrm{SE}} \leq T$, we find

$$
\begin{aligned}
\sum_{t \notin \mathcal{T}_{T+1}^{\mathrm{par}} \cup \mathcal{T}^{\mathrm{int}}} r(x_t, p_t\epsilon) \leq{} & 2|\mathcal{K}| \left( 4P\log(1/\delta) \times (128P)^2 + \frac{3\log(1/\delta) \times (32P)^2}{(50PL+28)\epsilon} + 1 \right) \\
& + 2(50PL+28)\epsilon T.
\end{aligned}
\tag{8}
$$

Combining Equations (7) and (8), we find that, conditioned on the event $\mathcal{E}^{\mathrm{SBIP}}$,

$$
\begin{aligned}
R_T \leq{} & \frac{2Pd\log\left(\frac{T+d}{d}\right)\left(P\sqrt{d\log\left(\frac{1+B^2T}{\delta}\right)} + A\right)^2}{\epsilon^2} + \frac{18P^3\log(1/\delta)}{\epsilon^2} \\
& + 2|\mathcal{K}| \left( 4P\log(1/\delta) \times (128P)^2 + \frac{3\log(1/\delta) \times (32P)^2}{(50PL+28)\epsilon} + 1 \right) + 2(50PL+28)\epsilon T.
\end{aligned}
$$

By setting $\epsilon = (d^2\log(T)^2/T)^{\frac{1}{3}}$ and $\delta = \left((38+16P\epsilon^{-1})(T+1)^2\right)^{-1}$, and using the definition of $\mathcal{K}$, we find that there exists a constant $\widetilde{C}'$ depending on $A$, $B$ $C$, and $L$ such that with probability $1 - 20T^{-2/3}$,

$$
R_T \leq \widetilde{C}'(d\log(T)T)^{2/3}.
$$

This concludes the proof of Theorem 2. Note that we can choose $\widetilde{C} = 461P^3 + 100PL + 56$, and the result holds provided that $T$ is large enough (namely, that $T \geq 2(38+16P)$, that $10 \leq P/\epsilon$, that $P^2 \times \frac{16}{3}d\log(T) \geq A$, that $P^3\log(T) \geq 1$, that $B^2 + 1 \leq T$, and that $(T/\log(T)^2)^{1/3} \geq 331^2/102 \times P^2$).

## E  Proofs of auxiliary lemmas

### E.1  Proof of Lemma 1

**Lemma 1.** *Under Assumptions 1 and 2, the expected gain from trade for $x$ at price $p$ is given by*

$$
\mathrm{EGFT}(x,p) = I(\delta^b)F^s(\delta^s) + J(\delta^s)D^b(\delta^b),
\tag{4}
$$

*where $\delta^s := p - x_t^\top \theta^s$, $\delta^b := p - x_t^\top \theta^b$, $I(\delta) := \int_\delta^C D^b(u)du$, and $J(\delta) := \int_{-C}^\delta F^s(u)du$.*

*Proof.* The proof relies on the characterization of the gain from trade given by Lemma 4.1 in Cesa-Bianchi et al. (2021). Recall that, under Assumption 1, we have $\xi_t^s \in [-C, C]$, and $\xi_t^b \in [-C, C]$. Then, denoting $f^s$ and $f^b$ the densities of $\xi_t^s$ and $\xi_t^b$, we have

$$\text{EGFT}(x, p)$$
$$= \int_{(s,b)\in[x^\top\theta^s - C, x^\top\theta^s + C]\times[x^\top\theta^b - C, x^\top\theta^b + C]} (b - s)\mathbb{I}\{s \leq p \leq b\}f^b(x^\top\theta^b - s)f^s(x^\top\theta^s - s)\,ds\,db.$$

Let $\tilde{f}^b : b \mapsto f^b(x^\top\theta^b - s)$ and $\tilde{f}^s : s \mapsto f^s(x^\top\theta^s - s)$ be the densities of $s_t$ and $b_t$ conditionnally on $x_t = x$, and note that outside of $[x^\top\theta^b - C, x^\top\theta^b + C] \times [x^\top\theta^s - C, x^\top\theta^s + C]$, $\tilde{f}^s(s)\tilde{f}^b(b) = 0$. With these notation, we have

$$\text{EGFT}(x, p) = \int_{(s,b)\in[-P,P]^2} (b - s)\mathbb{I}\{s \leq p \leq b\}\tilde{f}^b(b)\tilde{f}^s(s)\,ds\,db$$

$$= \int_{(s,b)\in[-P,p]\times[p,P]} \left(\int_{-P}^b du - \int_{-P}^s du\right) \tilde{f}^s(s)\tilde{f}^b(b)\,ds\,db.$$

Thus,

$$\text{EGFT}(x, p) = \int_{[-P,P]} \left(\int_{(s,b)\in[-P,p]\times[p\vee u,P]} \tilde{f}^s(s)\tilde{f}^b(b)\,ds\,db\right) du$$

$$- \int_{[-P,p]} \left(\int_{(s,b)\in[u,p]\times[p,P]} \tilde{f}^s(s)\tilde{f}^b(b)\,ds\,db\right) du.$$

This yields

$$\text{EGFT}(x, p) = \int_{[-P,p]} \left(\int_{(s,b)\in[-P,p]\times[p,P]} \tilde{f}^s(s)\tilde{f}^b(b)\,ds\,db\right) du$$

$$+ \int_{[p,P]} \left(\int_{(s,b)\in[-P,p]\times[u,P]} \tilde{f}^s(s)\tilde{f}^b(b)\,ds\,db\right) du$$

$$- \int_{[-P,p]} \left(\int_{(s,b)\in[u,p]\times[p,P]} \tilde{f}^s(s)\tilde{f}^b(b)\,ds\,db\right) du.$$

This, in turn, implies that

$$\text{EGFT}(x, p) = \int_{[-P,p]} \left(\int_{(s,b)\in[-P,u]\times[p,P]} \tilde{f}^s(s)\tilde{f}^b(b)\,ds\,db\right) du$$

$$+ \int_{[p,P]} \left(\int_{(s,b)\in[-P,p]\times[u,P]} \tilde{f}^s(s)\tilde{f}^b(b)\,ds\,db\right) du$$

$$= \int_{[p,P]} \tilde{f}^b(b)\,dd \int_{[-P,p]} \int_{[-P,u]} \tilde{f}^s(s)\,ds\,du$$

$$+ \int_{[-P,p]} \tilde{f}^s(s)\,ds \int_{[p,P]} \int_{[u,P]} \tilde{f}^b(b)\,db\,du.$$

By using the change in variables $\xi^s = s - x^\top\theta^s$, $u^s = u - x^\top\theta^s$, $\xi^b = b - x^\top\theta^b$, and $u^b = u - x^\top\theta^b$, and the definition of $\tilde{f}^s$ and $\tilde{f}^b$, we get

$$\text{EGFT}(x, p) = \int_{[p-x^\top\theta^b, P-x^\top\theta^b]} f(\xi^b)\,dd \int_{[-P,p]} \int_{[-P-x^\top\theta^s, u-x^\top\theta^s]} f(\xi^s)\,d\xi^s\,du$$

$$+ \int_{[-P-x^\top\theta^s, p-x^\top\theta^s]} f(\xi^s)\,d\xi^s \int_{[p,P]} \int_{[u-x^\top\theta^b, P-x^\top\theta^b]} f(\xi^b)\,d\xi^b\,du$$

$$= \int_{[p-x^\top\theta^b, P-x^\top\theta^b]} f(\xi^b)\,d\xi^b \int_{[-P-x^\top\theta^s, p-x^\top\theta^s]} \int_{[-P-x^\top\theta^s, u^s]} f(\xi^s)\,d\xi^s\,du^s$$

$$+ \int_{[-P-x^\top\theta^s, p-x^\top\theta^s]} f(\xi^s)\,d\xi^s \int_{[p-x^\top\theta^b, P-x^\top\theta^b]} \int_{[u^b, P-x^\top\theta^b]} f(\xi^b)\,d\xi^b\,du^b.$$

Now, under Assumption 1, $f^b$ and $f^s$ are null outside of $[-C, C]$. Moreover, $P - x^\top \theta^b \geq C$ and $-P - x^\top \theta^s \leq -C$. Thus,

$$
\begin{aligned}
\text{EGFT}(x, p) &= \int_{[p-x^\top\theta^b, C]} f(\xi^b) \, \mathrm{d}d \int_{[-C, p-x^\top\theta^s]} \int_{[-C, u^s]} f(\xi^s) \, \mathrm{d}\xi^s \, \mathrm{d}u^s \\
&\quad + \int_{[-C, p-x^\top\theta^s]} f(\xi^s) \, \mathrm{d}\xi^s \int_{[p-x^\top\theta^b, C]} \int_{[u^b, C]} f(\xi^b) \, \mathrm{d}\xi^b \, \mathrm{d}u^b \\
&= D^b(p - x^\top\theta^b) \int_{-C}^{p-x^\top\theta^s} F^s(u^s) \, \mathrm{d}u^s + F^s(p - x^\top\theta^s) \int_{p-x^\top\theta^b}^{C} D^b(u^b) \, \mathrm{d}u^b.
\end{aligned}
$$

Using the definition of $\delta^s = p - x^\top\theta^s$ and $\delta^b = p - x^\top\theta^b$, we obtain the desired result. □

### E.2 PROOF OF LEMMA 4

**Lemma 4.** *Almost surely, the length of exploration phase $\mathcal{T}_{T+1}^{\text{par}}$ is bounded as*

$$
|\mathcal{T}_{T+1}^{\text{par}}| \leq \frac{d \log\left(\frac{T+d}{d}\right)}{\mu^2}.
$$

*Proof.* The elliptical potential Lemma (see, *e.g.,* Proposition 1 in Carpentier et al. (2020)) implies that almost surely,

$$
\sum_{t \in \mathcal{T}_{T+1}^{\text{par}}} \|x_t\|_{\left(\sum_{l \in \mathcal{T}_t^{\text{par}}} x_l x_l^\top + \mathbf{I}_d\right)^{-1}} \leq \sqrt{|\mathcal{T}_{T+1}^{\text{par}}| d \log\left(\frac{|\mathcal{T}_{T+1}^{\text{par}}| + d}{d}\right)}.
$$

Since for all $t \in \mathcal{T}_{T+1}^{\text{par}}$, $\|x_t\|_{\left(\sum_{l \in \mathcal{T}_t^{\text{par}}} x_l x_l^\top + \mathbf{I}_d\right)^{-1}} \geq \mu$, this implies that

$$
|\mathcal{T}_{T+1}^{\text{par}}| \mu \leq \sqrt{|\mathcal{T}_{T+1}^{\text{par}}| d \log\left(\frac{|\mathcal{T}_{T+1}^{\text{par}}| + d}{d}\right)}.
$$

Now, almost surely, $|\mathcal{T}_{T+1}^{\text{par}}| \leq T$. This implies in particular that

$$
|\mathcal{T}_{T+1}^{\text{par}}| \leq \frac{d \log\left(\frac{T+d}{d}\right)}{\mu^2},
$$

which concludes the proof. □

### E.3 PROOF OF LEMMA 5

**Lemma 5.** *For the choice* $\mu = \epsilon \left( P\sqrt{d \log\left(\frac{1+B^2 T}{\delta}\right)} + A \right)^{-1}$, $\mathrm{T}^{\text{int}} = 8P^2 \log(1/\delta)/\epsilon^2$, *and* $\mathrm{T}^{\text{FD}} = 2\log(1/\delta)\epsilon^{-2}$, *it holds that*

$$
\mathbb{P}\left(\mathcal{E}^{\text{EOC}}\right) \geq 1 - 2\delta - 8\delta(2K + 1).
$$

*Proof.* The Lemma is a consequence of the following two auxiliary results, bounding respectively the error for estimating the parameters $\theta^s$ and $\theta^b$, and the integrals $I$ and $J$.

**Lemma 11.** *Let $\mathcal{E}_1$ be the event :*

$$
\mathcal{E}_1 := \left\{ \forall t \notin \mathcal{T}_{T+1}^{\text{par}}, \quad \left| x_t^\top (\widehat{\theta}_t^b - \theta^b) \right| \leq \epsilon \quad \text{and} \quad \left| x_t^\top (\widehat{\theta}_t^s - \theta^s) \right| \leq \epsilon \right\}.
$$

*Then, for the choice $\mu = \epsilon \left( P\left(d \log\left(\frac{1+B^2 T}{\delta}\right)\right)^{1/2} + A \right)^{-1}$, we have $\mathbb{P}(\mathcal{E}_1) \geq 1 - 2\delta$.*

The proof of the above lemma is deferred to Appendix E.9.

**Lemma 12.** *Let $\mathcal{E}_2$ be the event:*

$$\mathcal{E}_2 := \left\{ |\mathcal{T}^{\text{int}}| < \mathrm{T}^{\text{int}} \text{ or } \forall k \in \mathcal{K} \quad \left| \widehat{I}^k - I(k\epsilon) \right| \leq 2\epsilon \text{ and } \left| \widehat{J}^k - J(k\epsilon) \right| \leq 2\epsilon \right\} \cap \mathcal{E}_1.$$

*Then, for the choice $\mu = \epsilon \left( P \left( d \log \left( \frac{1+B^2 T}{\delta} \right) \right)^{1/2} + A \right)^{-1}$ and $\mathrm{T}^{\text{int}} = 8P^2 \log(1/\delta)/\epsilon^2$, we have*

$$\mathbb{P}(\mathcal{E}_2) \geq 1 - 2\delta - 4\delta(2K+1).$$

The proof of the above lemma is deferred to Appendix E.10.

Next, we control the error $|\widehat{F}^k - F^s(k\epsilon)|$ uniformly for $k \in \mathcal{K}$. The results for $|\widehat{D}^k - D^b(k\epsilon)|$ follow from similar arguments. To do so, we rely on the following well known result (for the sake of completeness, we provide a proof in Appendix E.11).

**Lemma 13.** *Let $(y_t)_{t \geq 1}$ be a sequence of random variables adapted for a filtration $\mathcal{F}_t$, such that $y_t - \mathbb{E}[y_t|\mathcal{F}_{t-1}] \in [m, M]$. Assume that for $t \in \mathbb{N}_*$, $\iota_t \in \{0,1\}$ is $\mathcal{F}_{t-1}$-measurable, and define $N_t := \sum_{l \leq t} \iota_l$, and $\widehat{\mu}_t := N_t^{-1} \sum_{l \leq t} \iota_l (y_l - \mathbb{E}[y_l|\mathcal{F}_{l-1}])$ if $N_t \geq 1$. Then, for any $t \in \mathbb{N}_*$ and $\delta \in (0,1)$,*

$$\mathbb{P}\left( N_t = 0 \text{ or } |\widehat{\mu}_t| \leq (M-m)\sqrt{\frac{\log(1/\delta)}{2N_t}} \right) \geq 1 - 2t\delta.$$

*Moreover, for any $t > 0$ and $\delta \in (0,1)$,*

$$\mathbb{P}\left( N_t = t \text{ and } |\widehat{\mu}_t| \geq (M-m)\sqrt{\frac{\log(1/\delta)}{2N_t}} \right) \leq 1 - 2\delta.$$

For $t \leq T$, we define $\iota_t := \mathbb{I}\{t \in \mathcal{T}_k^F\}$, $y_t := \mathbb{I}\{s_t \leq p_t\}$, and we observe that for $\mathcal{F}_t = \sigma\left((x_1, \ldots, x_{t+1}, (\mathbb{I}\{s_1 \leq p_1\}, \mathbb{I}\{b_1 \leq p_1\}), \ldots, (\mathbb{I}\{s_t \leq p_t\}, \mathbb{I}\{b_t \leq p_t\}))\right)$, $\iota_t$ is $\mathcal{F}_{t-1}$-measurable, and $y_t$ is $\mathcal{F}_t$ adapted. Moreover,

$$\iota_t \mathbb{E}[y_t \mid \mathcal{F}_{t-1}] = \iota_t \mathbb{P}\left( x_t^\top \theta^s + \xi_t^s \leq x_t^\top \widehat{\theta}_t^s + k\epsilon \right)$$
$$= \iota_t F^s \left( x_t^\top \left( \widehat{\theta}_t^s - \theta^s \right) + k\epsilon \right).$$

Using Lemma 13, we find that with probability $1 - 2\delta$, either $|\mathcal{T}_k^F| < \mathrm{T}^{\text{FD}}$, or

$$\left| \widehat{F}_t^k - \frac{\sum_{s \leq t} \iota_t F^s \left( x_t^\top \left( \widehat{\theta}_t^s - \theta^s \right) + k\epsilon \right)}{\mathrm{T}^{\text{FD}}} \right| \leq \sqrt{\frac{2\log(1/\delta)}{\mathrm{T}^{\text{FD}}}}.$$

Moreover, on the event $\mathcal{E}_1$, for all $t \notin \mathcal{T}^{\text{par}}$, $|x_t^\top(\widehat{\theta}_t^s - \theta^s)| \leq \epsilon$. Using the fact that $F^s$ is $L$-Lipschitz, we find that

$$\left| F^s \left( x_t^\top \left( \widehat{\theta}_t^s - \theta^s \right) + k\epsilon \right) - F^s(k\epsilon) \right| \leq L\epsilon.$$

Thus, with probability $1 - 2\delta$, either $|\mathcal{T}_k^F| < \mathrm{T}^{\text{FD}}$, or

$$\left| \widehat{F}_t^k - F^s(k\epsilon) \right| \leq \sqrt{\frac{2\log(1/\delta)}{\mathrm{T}^{\text{FD}}}} + L\epsilon.$$

Using Lemma 11 and 12, along with a union bound over $k \in \mathcal{K}$ yields the desired result for the choice $\mathrm{T}^{\text{FD}} = 2\log(1/\delta)\epsilon^{-2}$. $\qquad\square$

### E.4   PROOF OF LEMMA 6

**Lemma 6.** *On the event $\mathcal{E}^{\text{EOC}}$, for all $t \in \mathcal{T}^{\text{C}}$, and all $(k, k') \in \mathcal{A}_t$,*

$$\left| \widehat{I}^{k'} \widehat{F}^k + \widehat{J}^k \widehat{D}^{k'} - \text{EGFT}(x_t, x_t^\top \widehat{\theta}_t^s + k\epsilon) \right| \leq (10PL + 4P + 7)\epsilon.$$

*Proof.* The proof relies on the following result :

**Lemma 14.** *On the event $\mathcal{E}^{\text{ETC}}$, for all $(k, k') \in \mathcal{A}_t$, and $p = x_t^\top \widehat{\theta}_t^s + k\epsilon$, we have*

$$\left| \text{EGFT}(x_t, p) - \left( \widehat{I}^{k'} \widehat{F}^k + \widehat{J}^k \widehat{D}^{k'} \right) \right| \leq 6PL\epsilon + 3\epsilon + 2P \left( |\widehat{\Delta}F| + |\widehat{\Delta}D| \right) + |\widehat{\Delta}I| + |\widehat{\Delta}J|$$

*where we define $\widehat{\Delta}I = I(k'\epsilon) - \widehat{I}^{k'}$, $\widehat{\Delta}J = J(k\epsilon) - \widehat{J}^k$, $\widehat{\Delta}F = F(k\epsilon) - \widehat{F}^k$, $\widehat{\Delta}D = D(k'\epsilon) - \widehat{D}^{k'}$.*

The proof of this intermediate result is in Appendix E.12.

To conclude the proof of Lemma 6, it remains to bound the gaps $\widehat{\Delta}F$, $\widehat{\Delta}D$, $\widehat{\Delta}I$, and $\widehat{\Delta}J$ on the event $\mathcal{E}^{\text{ETC}}$. By definition of the event $\mathcal{E}^{\text{ETC}}$, on this event $\widehat{\Delta}I \leq 2\epsilon$, $\widehat{\Delta}D \leq (L+1)\epsilon$, $\widehat{\Delta}J \leq 2\epsilon$ and $\widehat{\Delta}F \leq (L+1)\epsilon$. Thus, on the event $\mathcal{E}^{\text{ETC}}$,

$$\left| \text{EGFT}(x_t, p) - \left( \widehat{I}^{k'} \widehat{F}^k + \widehat{J}^k \widehat{D}^{k'} \right) \right| \leq 6PL\epsilon + 3\epsilon + 4P(L+1)\epsilon + 4\epsilon.$$

This concludes the proof. □

### E.5   PROOF OF LEMMA 7

**Lemma 7.** *On the event $\mathcal{E}^{\text{EOC}}$, we have that*

$$\left| \max_p \text{EGFT}(x_t, p) - \text{EGFT}(x_t, x_t^\top \widehat{\theta}_t^s + k_t^* \epsilon) \right| \leq 2LP\epsilon.$$

*Proof.* By definition of $\mathcal{A}_t$,

$$\sup_{p \in [-P, P]} \text{EGFT}(x_t, p) \geq \text{EGFT}(x_t, x_t^\top \widehat{\theta}_t^s + k_t^* \epsilon).$$

Since the mapping $p \mapsto \text{EGFT}(x, p)$ is continuous, we can define

$$p_t^* \in \arg\max_p \text{EGFT}(x_t, p), \quad \tilde{k}_t = \left\lfloor \frac{p_t^* - x_t^\top \widehat{\theta}_t^s}{\epsilon} \right\rfloor, \quad \text{and} \quad \tilde{k}'_t = \left\lfloor \frac{x_t^\top \left( \widehat{\theta}_t^s - \widehat{\theta}_t^b \right) + \tilde{k}_t}{\epsilon} \right\rfloor.$$

We now show that $(\tilde{k}_t, \tilde{k}'_t) \in \mathcal{A}_t$. By Lemma 1, we have that $p_t^* - x_t^\top \theta^s \geq -C$, and $p_t^* - x_t^\top \theta^b \leq C$ (otherwise $\text{EGFT}(x_t, p_t^*) = 0$). This implies that $p_t^* - x_t^\top \theta^s \in [-2P, 2P]$, and similarly that $p_t^* - x_t^\top \theta^b \in [-2P, 2P]$.

On the one hand, on the event $\mathcal{E}^{\text{ETC}}$, $\left| x_t^\top \left( \theta^s - \widehat{\theta}_t^s \right) \right| \leq \epsilon$, so $p_t^* - x_t^\top \widehat{\theta}_t^s \in [-2P - \epsilon, 2P + \epsilon]$. Thus, $\tilde{k}_t \in \mathcal{K}$. On the other hand,

$$x_t^\top \left( \widehat{\theta}_t^s - \widehat{\theta}_t^b \right) + \tilde{k}_t \epsilon = \left( x_t^\top \widehat{\theta}_t^s + \tilde{k}_t \epsilon - p_t^* \right) + (p_t^* - x_t^\top \theta^b) + x_t^\top \left( \theta^b - \widehat{\theta}_t^b \right).$$

Then, on the event $\mathcal{E}^{\text{ETC}}$,

$$x_t^\top \left( \widehat{\theta}_t^s - \widehat{\theta}_t^b \right) + \tilde{k}_t \epsilon \in [-2P - 2\epsilon, 2P + 2\epsilon],$$

, so $k'_t \in [-2P - 3\epsilon, 2P + 3\epsilon]$. Therefore, $\tilde{k}'_t \in \mathcal{K}$. This implies that $(\tilde{k}_t, \tilde{k}'_t) \in \mathcal{A}_t$, so

$$\text{EGFT}(x_t, x_t^\top \widehat{\theta}_t^s + k_t^* \epsilon) \geq \text{EGFT}(x_t, x_t^\top \widehat{\theta}_t^s + \tilde{k}_t \epsilon).$$

Finally, Lemma 1 and Assumption 2 imply that the function $p \mapsto \text{EGFT}(x_t, p)$ is $2LP$-Lipschitz continuous. This, in turn, implies that

$$\text{EGFT}(x_t, p_t^*) - \text{EGFT}(x_t, x_t^\top \widehat{\theta}_t^s + \tilde{k}_t \epsilon) \leq 2LP\epsilon.$$

This proves the statement. □

### E.6 Proof of Lemma 8

**Lemma 8.** *For the choice* $\mu = \epsilon \left( P \left( d \log \left( \frac{1 + B^2 T}{\delta} \right) \right)^{1/2} + A \right)^{-1}$ *and* $\mathrm{T}^{\mathrm{int}} = 8 P^2 \log(1/\delta) / \epsilon^2$*, it holds that*

$$\mathbb{P} \left( \mathcal{E}^{\mathrm{SBIP}} \right) \geq 1 - 2\delta - 4\delta(2K + 1) - 4\delta(2K + 1)T.$$

*Proof.* The proof relies on Lemma 11 and 12. On top of these results, we have to control the error $|\widehat{F}_t^k - F^s(k\epsilon)|$ uniformly for $k \in \mathcal{K}$. To do so, we rely on Lemma 13. For $t \leq T$, we define $\iota_t = \mathbb{I}\{t \notin \mathcal{T}^{\mathrm{int}} \cup \mathcal{T}_{T+1}^{\mathrm{par}}$ and $k_t = k\}$, $y_t = \mathbb{I}\{s_t \leq p_t\}$, and we note that for $\mathcal{F}_t = \sigma \left( (x_1, \ldots, x_{t+1}, (\mathbb{I}\{s_1 \leq p_1\}, \mathbb{I}\{b_1 \leq p_1\}), \ldots, (\mathbb{I}\{s_t \leq p_t\}, \mathbb{I}\{b_t \leq p_t\})) \right)$, $\iota_t$ is $\mathcal{F}_{t-1}$-measurable, and $y_t$ is $\mathcal{F}_t$ adapted. Moreover,

$$\iota_t \mathbb{E}\left[y_t \mid \mathcal{F}_{t-1}\right] = \iota_t \mathbb{P}\left( x_t^\top \theta^s + \xi_t^s \leq x_t^\top \widehat{\theta}_t^s + k\epsilon \right)$$
$$= \iota_t F^s \left( x_t^\top \left( \widehat{\theta}_t^s - \theta^s \right) + k\epsilon \right).$$

Using Lemma 13, we find that with probability $1 - 2\delta t$, either $N_t^{s,k} = 0$ or

$$\left| \widehat{F}_t^k - \frac{\sum_{s \leq t} \iota_t F^s \left( x_t^\top \left( \widehat{\theta}_t^s - \theta^s \right) + k\epsilon \right)}{N_t^{s,k}} \right| \leq \sqrt{\frac{2 \log(1/\delta)}{N_t^{s,k}}}$$

(where we recall that we adopt the convention $1/0 = \infty$). Moreover, on the event $\mathcal{E}_1$, for all $t \notin \mathcal{T}^{\mathrm{par}}$, $|x_t^\top(\widehat{\theta}_t^s - \theta^s)| \leq \epsilon$. Since $F^s$ is $L$-Lipschitz, this implies

$$\left| F^s \left( x_t^\top \left( \widehat{\theta}_t^s - \theta^s \right) + k\epsilon \right) - F^s (k\epsilon) \right| \leq L\epsilon.$$

Thus, with probability $1 - 2\delta t$,

$$\left| \widehat{F}_t^k - F^s(k\epsilon) \right| \leq \sqrt{\frac{2 \log(1/\delta)}{N_t^{s,k}}} + L\epsilon.$$

Using a union bound over $k \in \mathcal{K}$ and $t \notin \mathcal{T}^{\mathrm{int}} \cup \mathcal{T}_{T+1}^{\mathrm{par}}$ yields the desired result.

Next, we can bound $|\widehat{D}_t^k - D^b(k\epsilon)|$ using similar arguments. In particular, we define $\iota_t = \mathbb{I}\{t \notin \mathcal{T}^{\mathrm{int}} \cup \mathcal{T}_{T+1}^{\mathrm{par}}$ and $k_t' = k\}$, $y_t = \mathbb{I}\{b_t \geq p_t\}$ and note that

$$\iota_t \mathbb{E}\left[y_t \mid \mathcal{F}_{t-1}\right] = \iota_t \mathbb{P}\left( x_t^\top \theta^b + \xi_t^b \geq x_t^\top \widehat{\theta}_t^s + k_t \epsilon \mid \mathcal{F}_{t-1} \right)$$
$$= \iota_t D^b \left( x_t^\top \left( \widehat{\theta}_t^b - \theta^b \right) + k\epsilon + \epsilon \left( \frac{x_t^\top \widehat{\theta}_t^s - x_t^\top \widehat{\theta}_t^b + k_t \epsilon}{\epsilon} - k \right) \right).$$

By definition of $\mathcal{A}_t$, we have that if $\iota_t = 1$, then $\left| \frac{x_t^\top \widehat{\theta}_t^s - x_t^\top \widehat{\theta}_t^b + k_t \epsilon}{\epsilon} - k \right| \leq 1$. Using the Lipschitz continuity of $D^b$, this implies that, conditioned on the event $\mathcal{E}_1$,

$$\iota_t \left| \mathbb{E}\left[y_t \mid \mathcal{F}_{t-1}\right] - D^b(k\epsilon) \right| \leq 2L\epsilon.$$

The rest of the proof follows similarly. $\qquad \square$

### E.7 Proof of Lemma 9

**Lemma 9.** *Under the assumptions of Lemma 8 and conditioned on the event $\mathcal{E}^{\mathrm{SBIP}}$, we have that for all $t \notin \mathcal{T}_{T+1}^{\mathrm{par}} \cup \mathcal{T}^{\mathrm{int}}$, and all $(k, k') \in \mathcal{A}_t$;*

$$\mathrm{LCB}_t(k, k') \leq \mathrm{EGFT}(x_t, x_t^\top \widehat{\theta}_t^s + k\epsilon) \leq \mathrm{UCB}_t(k, k').$$

*Moreover, it holds that $(k_t^*, k_t'^*) \in \mathcal{K}_t$, where we recall that*

$$(k_t^*, k_t'^*) \in \underset{(k, k') \in \mathcal{A}_t}{\arg\max} \ \mathrm{EGFT}(x_t, x_t^\top \widehat{\theta}_t^s + k\epsilon).$$

*Proof.* Let us prove the first part of Lemma 9. By Lemma 14, and by definition of the event $\mathcal{E}^{\text{SBIP}}$, we have for all $(k, k') \in \mathcal{A}_t$, and all $t \notin \mathcal{T}_{T+1}^{\text{par}} \cup \mathcal{T}^{\text{int}}$,

$$\left| \text{EGFT}(x_t, p) - \left( \widehat{I}^{k'} \widehat{F}^k + \widehat{J}^k \widehat{D}^{k'} \right) \right|$$

$$\leq 6PL\epsilon + 3\epsilon + 2P \left( \sqrt{\frac{2\log(1/\delta)}{N_t^{s,k}}} + \sqrt{\frac{2\log(1/\delta)}{N_t^{b,k'}}} + 3L\epsilon \right) + 4\epsilon$$

$$\leq (12PL + 7)\epsilon + 2P \left( \sqrt{\frac{2\log(1/\delta)}{N_t^{s,k}}} + \sqrt{\frac{2\log(1/\delta)}{N_t^{b,k'}}} \right).$$

This concludes the first part of Lemma 9. The second claim follows immediately by noticing that by definition, $(k_t^*, k_t'^*) \in \mathcal{A}_t$, and that, for all $(k, k') \in \mathcal{A}_t$,

$$\text{LCB}_t(k, k') \leq \text{EGFT}(x_t, x_t^\top \widehat{\theta}_t^s + k\epsilon) \leq \text{EGFT}(x_t, x_t^\top \widehat{\theta}_t^s + k_t^* \epsilon) \leq \text{UCB}_t(k_t^*, k_t'^*).$$

This concludes the proof. $\qquad\square$

### E.8 Proof of Lemma 10

**Lemma 10.** *Conditioned on the event $\mathcal{E}^{\text{SBIP}}$, if $t \in \mathcal{T}_{s,k}^{\text{SE}}$, then the following condition holds*

$$r(x_t, p_t) \leq (50PL + 28)\epsilon + 16P \sqrt{\frac{2\log(1/\delta)}{N_t^{s,k}}}.$$

*Similarly, if $t \in \mathcal{T}_{b,k'}^{\text{SE}}$, then the following condition holds*

$$r(x_t, p_t) \leq (50PL + 28)\epsilon + 16P \sqrt{\frac{2\log(1/\delta)}{N_t^{b,k'}}}.$$

*Proof.* Assume that $t \in \mathcal{T}_{s,k}^{\text{SE}}$. Then, our choice of $k_t$ together with Lemma 9 ensures that, conditioned on the event $\mathcal{E}^{\text{SBIP}}$,

$$\text{UCB}_t(k_t, k_t') \geq \text{LCB}_t(k_t^*, k_t'^*).$$

This implies that

$$\text{LCB}_t(k_t, k_t') + (\text{UCB}_t(k_t, k_t') - \text{LCB}_t(k_t, k_t')) \geq$$
$$\text{UCB}_t(k_t^*, k_t'^*) + (\text{LCB}_t(k_t^*, k_t'^*) - \text{UCB}_t(k_t^*, k_t'^*)).$$

Using again Lemma 9, this implies that

$$\text{EGFT}(x_t, x_t^\top \widehat{\theta}^s + k_t \epsilon) + (\text{UCB}_t(k_t, k_t') - \text{LCB}_t(k_t, k_t')) + (\text{UCB}_t(k_t^*, k_t'^*) - \text{LCB}_t(k_t^*, k_t'^*))$$
$$\geq \text{EGFT}(x_t, x_t^\top \widehat{\theta}^s + k_t^* \epsilon).$$

Thus,

$$\text{EGFT}(x_t, x_t^\top \widehat{\theta}^s + k_t^* \epsilon) - \text{EGFT}(x_t, x_t^\top \widehat{\theta}^s + k_t \epsilon)$$
$$\leq (\text{UCB}_t(k_t, k_t') - \text{LCB}_t(k_t, k_t')) + (\text{UCB}_t(k_t^*, k_t'^*) - \text{LCB}_t(k_t^*, k_t'^*)).$$

Then, conditioned on the event $\mathcal{E}^{\text{SBIP}}$,

$$\text{UCB}_t(k_t, k_t') - \text{LCB}_t(k_t, k_t') \leq 2(12PL + 7)\epsilon + 4P \left( \sqrt{\frac{2\log(1/\delta)}{N_t^{s,k}}} + \sqrt{\frac{2\log(1/\delta)}{N_t^{b,k'}}} \right).$$

Since $t \in \mathcal{T}_{s,k}^{\text{SE}}$, we know that $N_t^{b,k_t'} \geq N_t^{s,k_t}$. Then, we have that

$$\text{UCB}_t(k_t, k_t') - \text{LCB}_t(k_t, k_t') \leq 2(12PL + 7)\epsilon + 8P \sqrt{\frac{2\log(1/\delta)}{N_t^{s,k}}}.$$

Similarly, since $t \in \mathcal{T}_{s,k}^{\text{SE}}$, we have that $N_t^{b,k_t'^*} \geq N_t^{s,k_t}$, and $N_t^{s,k_t^*} \geq N_t^{s,k_t}$, so we also have

$$\text{UCB}_t(k_t^*, k_t'^*) - \text{LCB}_t(k_t^*, k_t'^*) \leq 2(12PL + 7)\epsilon + 8P\sqrt{\frac{2\log(1/\delta)}{N_t^{s,k}}}.$$

Thus,

$$\text{EGFT}(x_t, x_t^\top \widehat{\theta}^s + k_t^* \epsilon) - \text{EGFT}(x_t, x_t^\top \widehat{\theta}^s + k_t \epsilon) \leq 4(12PL + 7)\epsilon + 16P\sqrt{\frac{2\log(1/\delta)}{N_t^{s,k}}}.$$

By Lemma 7, this implies

$$r(x_t, x_t^\top \widehat{\theta}^s + k_t \epsilon) \leq (50PL + 28)\epsilon + 16P\sqrt{\frac{2\log(1/\delta)}{N_t^{s,k}}}.$$

The proof of the second claim follows from similar arguments. $\square$

### E.9 PROOF OF LEMMA 11

**Lemma 11.** *Let $\mathcal{E}_1$ be the event :*

$$\mathcal{E}_1 := \left\{ \forall t \notin \mathcal{T}_{T+1}^{\text{par}}, \quad \left| x_t^\top (\widehat{\theta}_t^b - \theta^b) \right| \leq \epsilon \quad \text{and} \quad \left| x_t^\top (\widehat{\theta}_t^s - \theta^s) \right| \leq \epsilon \right\}.$$

*Then, for the choice $\mu = \epsilon \left( P \left( d \log \left( \frac{1 + B^2 T}{\delta} \right) \right)^{1/2} + A \right)^{-1}$, we have $\mathbb{P}(\mathcal{E}_1) \geq 1 - 2\delta$.*

*Proof.* Let us prove the bound $|x_t^\top (\widehat{\theta}_t^s - \theta^s)| \leq \epsilon$ with high probability; the bound on $|x_t^\top (\widehat{\theta}_t^b - \theta^b)|$ will follow from similar arguments.

We introduce the variables

$$\tilde{x}_t := x_t \mathbb{I}\left\{ t \in \mathcal{T}_{t+1}^{\text{par}} \right\} \quad \text{and} \quad \tilde{y}_t := 2P\, \mathbb{I}\left\{ t \in \mathcal{T}_{t+1}^{\text{par}} \right\} \left( \mathbb{I}\{p_t \leq s_t\} - \frac{1}{2} \right)$$

and the $\sigma$-algebra $\mathcal{F}_t = \sigma\left( (x_1, \ldots, x_{t+1}, (\mathbb{I}\{s_1 \leq p_1\}, \mathbb{I}\{b_1 \leq p_1\}), \ldots, (\mathbb{I}\{s_t \leq p_t\}, \mathbb{I}\{b_t \leq p_t\})) \right).$ With these notation, we have that $\tilde{x}_t$ and $\{t \in \mathcal{T}_{t+1}^{\text{par}}\}$ are $\mathcal{F}_t$-measurable. Moreover,

$$\mathbb{E}\left[ \tilde{y}_t | \mathcal{F}_{t-1} \right] = \mathbb{I}\left\{ t \in \mathcal{T}_{t+1}^{\text{par}} \right\} \cdot \left( 2P \int_{-P}^{P} \mathbb{P}\left[ u \leq s_t | \mathcal{F}_{t-1} \right] \frac{du}{2P} - P \right)$$

$$= \mathbb{I}\left\{ t \in \mathcal{T}_{t+1}^{\text{par}} \right\} \cdot \left( \int_{-P}^{P} \int_{-C}^{C} \mathbb{I}\left\{ u \leq x_t^\top \theta^s + \xi \right\} f^s(\xi)\, d\xi\, du - P \right)$$

$$= \mathbb{I}\left\{ t \in \mathcal{T}_{t+1}^{\text{par}} \right\} \cdot \left( \int_{-C}^{C} \int_{-P}^{\xi + x_t^\top \theta^s} du\, f^s(\xi)\, d\xi - P \right)$$

$$= \mathbb{I}\left\{ t \in \mathcal{T}_{t+1}^{\text{par}} \right\} \cdot \left( x_t^\top \theta^s + \int_{-C}^{C} \xi f^s(\xi)\, d\xi \right)$$

$$= \mathbb{I}\left\{ t \in \mathcal{T}_{t+1}^{\text{par}} \right\} x_t^\top \theta^s$$

where in the last equality we used that $\int_{-C}^{C} \xi f^s(\xi)\, d\xi = \mathbb{E}[\xi_t^s] = 0$. Thus, conditionally on $\mathcal{F}_{t-1}$, $\tilde{y}_t - \tilde{x}_t^\top \theta^s$ is centered and it belongs to $[-P, P]$, which implies that it is $P$-subgaussian. Now, for all $t \in \{1, \ldots, T\}$, we have

$$\widehat{\theta}_t^s = 2P \left( \sum_{l \in \mathcal{T}_{T+1}^{\text{par}}} x_l x_l^\top + \mathbf{I}_d \right)^{-1} \sum_{s \in \mathcal{T}_{T+1}^{\text{par}}} \left( \mathbb{I}\{p_t \geq s_t\} - \frac{1}{2} \right) x_l$$

$$= \left( \sum_{l < t} \tilde{x}_l \tilde{x}_l^\top + \mathbf{I}_d \right)^{-1} \sum_{l < t} \tilde{y}_l \tilde{x}_l.$$

Using the fact that, for all $t \leq 1$, $\|\tilde{x}_t\| \leq B$ and $\|\theta^s\| \leq A$, and applying Theorem 2 by Abbasi-Yadkori et al. (2011), we find that for all $t \geq 0$, with probability $1 - \delta$,

$$\|\widehat{\theta}_t^s - \theta^s\|_{\sum_{s<t} \tilde{x}_l \tilde{x}_l^\top + \mathbf{I}_d} \leq P\sqrt{d \log\left(\frac{1+B^2T}{\delta}\right)} + A.$$

Note that our definitions of $\tilde{x}_t$ and $\tilde{y}_t$ ensure that

$$\|\widehat{\theta}_t^s - \theta^s\|_{\sum_{s \in \mathcal{T}_{T+1}^{\text{par}}} x_l x_l^\top + \mathbf{I}_d} = \|\widehat{\theta}_t^s - \theta^s\|_{\sum_{s<t} \tilde{x}_l \tilde{x}_l^\top + \mathbf{I}_d}.$$

Moreover, for all $t$,

$$|x_t^\top(\widehat{\theta}_t^s - \theta^s)| \leq \|x_t^\top\|_{\left(\sum_{s \in \mathcal{T}_{T+1}^{\text{par}}} x_l x_l^\top + \mathbf{I}_d\right)^{-1}} \|\widehat{\theta}_t^s - \theta^s\|_{\left(\sum_{s \in \mathcal{T}_{T+1}^{\text{par}}} x_l x_l^\top + \mathbf{I}_d\right)}.$$

In particular, if $t \notin \mathcal{T}_{T+1}^{\text{par}}$, $\|x_t^\top\|_{\left(\sum_{s \in \mathcal{T}_{T+1}^{\text{par}}} x_l x_l^\top + \mathbf{I}_d\right)^{-1}} \leq \mu$, so with probability $1 - \delta$,

$$|x_t^\top(\widehat{\theta}_t^s - \theta^s)| \leq \mu\left(P\sqrt{d \log\left(\frac{1+B^2T}{\delta}\right)} + A\right).$$

For the choice $\mu = \epsilon\left(P\sqrt{d \log\left(\frac{1+B^2T}{\delta}\right)} + A\right)^{-1}$, this implies that

$$|x_t^\top(\widehat{\theta}_t^s - \theta^s)| \leq \epsilon,$$

which concludes the proof. $\qquad\square$

### E.10 PROOF OF LEMMA 12

**Lemma 12.** *Let $\mathcal{E}_2$ be the event:*

$$\mathcal{E}_2 := \left\{|\mathcal{T}^{\text{int}}| < \mathrm{T}^{\text{int}} \text{ or } \forall k \in \mathcal{K} \quad \left|\widehat{I}^k - I(k\epsilon)\right| \leq 2\epsilon \text{ and } \left|\widehat{J}^k - J(k\epsilon)\right| \leq 2\epsilon\right\} \cap \mathcal{E}_1.$$

*Then, for the choice $\mu = \epsilon\left(P\left(d \log\left(\frac{1+B^2T}{\delta}\right)\right)^{1/2} + A\right)^{-1}$ and $\mathrm{T}^{\text{int}} = 8P^2 \log(1/\delta)/\epsilon^2$, we have*

$$\mathbb{P}(\mathcal{E}_2) \geq 1 - 2\delta - 4\delta(2K+1).$$

*Proof.* We control the error $\left|\widehat{I}^k - I(k\epsilon)\right|$ uniformly for $k \leq K$; the result for $\left|\widehat{J}^k - J(k\epsilon)\right|$ can be proved analogously.

For $k \leq K$, and $t \leq T$, let us define $\iota_t = \mathbb{I}\{t \in \mathcal{T}^{\text{int}}\}$, $y_t = 2P\mathbb{I}\left\{k\epsilon + x_t^\top\widehat{\theta}_t^b \leq p_l \leq b_t\right\}$ and note that for $\mathcal{F}_t = \sigma\left((x_1, \ldots, x_{t+1}, (\mathbb{I}\{s_1 \leq p_1\}, \mathbb{I}\{b_1 \leq p_1\}), \ldots, (\mathbb{I}\{s_t \leq p_t\}, \mathbb{I}\{b_t \leq p_t\}))\right)$, $\iota_t$ is $\mathcal{F}_{t-1}$-measurable, and $y_t$ is $\mathcal{F}_t$-adapted. Moreover,

$$\mathbb{E}[y_t|\mathcal{F}_{t-1}] = 2P\mathbb{P}\left[k\epsilon + x_t^\top\widehat{\theta}_t^b \leq p_t \leq b_t \,\middle|\, \mathcal{F}_{t-1}\right]$$

$$= 2P\int_{k\epsilon+x_t^\top\widehat{\theta}_t^b}^{P}\int_{-C}^{C} \mathbb{I}\{x_t^\top\theta^b + \xi \geq u\} f^b(\xi) \,\mathrm{d}\xi\frac{\mathrm{d}u}{2P}$$

$$= \int_{k\epsilon+x_t^\top\widehat{\theta}_t^b}^{P} D^b\left(u - x_t^\top\theta^b\right)\mathrm{d}u.$$

Using the change in variables $u' = u - x_t^\top\theta^b$, this implies that

$$\mathbb{E}[y_t|\mathcal{F}_{t-1}] = \int_{k\epsilon+x_t^\top\widehat{\theta}_t^b-x_t^\top\theta^b}^{P-x_t^\top\theta^b} D^b(u')\,\mathrm{d}u'.$$

Moreover, under Assumptions 1 and 2, $P - x_t^\top \theta^b \geq C$ and, for $u' \in [C, P - x_t^\top \theta^b]$, $D^b(u') = 0$. Thus,

$$
\mathbb{E}\left[y_t | \mathcal{F}_{t-1}\right] = \int_{k\epsilon + x_t^\top \widehat{\theta}_t^b - x_t^\top \theta^b}^{C} D^b(u') \, \mathrm{d}u'
$$
$$
= I(k\epsilon + x_t^\top \widehat{\theta}_t^b - x_t^\top \theta^b).
$$

Finally, note that $y_t - \mathbb{E}\left[y_t | \mathcal{F}_{t-1}\right]$ is in $[-2P, 2P]$. Then, using Lemma 13, we find that

$$
\mathbb{P}\left( |\mathcal{T}^{\mathrm{int}}| = \mathrm{T}^{\mathrm{int}} \text{ and } \left| \widehat{I}^k - \frac{\sum_{t \in \mathcal{T}^{\mathrm{int}}} I\left(k\epsilon + x_t^\top \left(\widehat{\theta}_t^b - \theta^b\right)\right)}{\mathrm{T}^{\mathrm{int}}} \right| \geq 4P\sqrt{\frac{\log(1/\delta)}{2\mathrm{T}^{\mathrm{int}}}} \right) \leq 2\delta.
$$

To conclude our proof, note that on the event $\mathcal{E}_1$, for all $t \in \mathcal{T}^{\mathrm{int}}$, $|x_t^\top(\widehat{\theta}_t^b - \theta^b)| \leq \epsilon$, so with probability $1 - 2\delta$, either $|\mathcal{T}^{\mathrm{int}}| < \mathrm{T}^{\mathrm{int}}$ or

$$
\left| I\left(k\epsilon + x_t^\top \left(\widehat{\theta}_t^b - \theta^b\right)\right) - I(k\epsilon) \right| = \left| \int_{k\epsilon}^{k\epsilon + x_t^\top \left(\widehat{\theta}_t^b - \theta^b\right)} \left(1 - F^b(\lambda)\right) d\lambda \right|
$$
$$
\leq \left| \int_{k\epsilon}^{k\epsilon + x_t^\top \left(\widehat{\theta}_t^b - \theta^b\right)} d\lambda \right|
$$
$$
\leq \left| x_t \left(\widehat{\theta}_t^b - \theta^b\right) \right|
$$
$$
\leq \epsilon.
$$

Using the same reasoning to control the error in estimating $J$, taking a union bound for $k \in \mathcal{K}$, and using Lemma 11, we find that on an event $\mathcal{E}_2 \subset \mathcal{E}_1$ of probability larger than $1 - 2\delta - 4\delta(2K + 1)$, either $|\mathcal{T}^{\mathrm{int}}| < \mathrm{T}^{\mathrm{int}}$ or

$$
\left| \widehat{I}^k - I(k\epsilon) \right| \leq 4P\sqrt{\frac{\log(1/\delta)}{2\mathrm{T}^{\mathrm{int}}}} + \epsilon
$$

and

$$
\left| \widehat{J}^k - J(k\epsilon) \right| \leq 4P\sqrt{\frac{\log(1/\delta)}{2\mathrm{T}^{\mathrm{int}}}} + \epsilon
$$

simultaneously for all $k \in \mathcal{K}$. For the choice $\mathrm{T}^{\mathrm{int}} = 8P^2 \log(1/\delta)/\epsilon^2$, we obtain the desired result. $\square$

### E.11 PROOF OF LEMMA 13

**Lemma 13.** *Let $(y_t)_{t \geq 1}$ be a sequence of random variables adapted for a filtration $\mathcal{F}_t$, such that $y_t - \mathbb{E}\left[y_t | \mathcal{F}_{t-1}\right] \in [m, M]$. Assume that for $t \in \mathbb{N}_*$, $\iota_t \in \{0, 1\}$ is $\mathcal{F}_{t-1}$-measurable, and define $N_t := \sum_{l \leq t} \iota_l$, and $\widehat{\mu}_t := N_t^{-1} \sum_{l \leq t} \iota_l(y_l - \mathbb{E}\left[y_l | \mathcal{F}_{l-1}\right])$ if $N_t \geq 1$. Then, for any $t \in \mathbb{N}_*$ and $\delta \in (0, 1)$,*

$$
\mathbb{P}\left( N_t = 0 \text{ or } |\widehat{\mu}_t| \leq (M - m)\sqrt{\frac{\log(1/\delta)}{2N_t}} \right) \geq 1 - 2t\delta.
$$

*Moreover, for any $t > 0$ and $\delta \in (0, 1)$,*

$$
\mathbb{P}\left( N_t = t \text{ and } |\widehat{\mu}_t| \geq (M - m)\sqrt{\frac{\log(1/\delta)}{2N_t}} \right) \leq 1 - 2\delta.
$$

*Proof.* Let us define $Z_t := \sum_{l \leq t} \iota_l(y_l - \mathbb{E}\left[y_l | \mathcal{F}_{l-1}\right])$, and for all $x \in \mathbb{R}$ let $M_t := \exp\left\{x Z_t - \frac{1}{8}x^2(M - m)^2 N_t\right\}$. We begin by showing that $M_t$ is a super-martingale. Indeed,

we have that

$$\mathbb{E}\left[\exp\{x\,\iota_t(y_t - \mathbb{E}\left[y_t|\mathcal{F}_{t-1}\right])\} \mid \mathcal{F}_{t-1}\right] = \mathbb{E}\left[\iota_t \exp\{x(y_t - \mathbb{E}\left[y_t|\mathcal{F}_{t-1}\right])\} + (1 - \iota_t) \mid \mathcal{F}_{t-1}\right]$$

$$\le \iota_t \exp\left\{\frac{x^2(M-m)^2}{8}\right\} + (1 - \iota_t)$$

$$\le \exp\left\{\frac{x^2(M-m)^2\iota_t}{8}\right\},$$

where we use the fact that $(y_t - \mathbb{E}\left[y_t \mid \mathcal{F}_{t-1}\right])$ is bounded in $[m, M]$ together with the conditional version of Hoeffding's Lemma. Noticing that

$$M_t = M_{t-1} \exp\left\{x\,\iota_t(y_t - \mathbb{E}\left[y_t|\mathcal{F}_{t-1}\right]) - \frac{x^2(M-m)^2\iota_t}{8}\right\},$$

this proves that $M_t$ is a super-martingale, and so $\mathbb{E}\left[M_t\right] \le \mathbb{E}\left[M_0\right] = 1$.

Now, for all $\epsilon > 0$ and all $l \in \mathbb{N}$, and all $x > 0$, by a Markov-Chernoff argument,

$$\mathbb{P}\left(Z_t \ge \epsilon \text{ and } N_t = l\right) = \mathbb{P}\left(\mathbb{I}\left\{N_t = l\right\}e^{xZ_t} \ge e^{\epsilon x}\right)$$

$$\le e^{-\epsilon x}\mathbb{E}\left(\mathbb{I}\left\{N_t = l\right\} \cdot e^{xZ_t}\right)$$

$$= e^{-\epsilon x + \frac{x^2(M-m)^2 l}{8}}\mathbb{E}\left(\mathbb{I}\left\{N_t = l\right\} \cdot e^{xZ_t - \frac{x^2(M-m)^2 l}{8}}\right).$$

Using the previous result, we have that

$$\mathbb{E}\left(\mathbb{I}\left\{N_t = l\right\} \cdot e^{xZ_t - \frac{x^2(M-m)^2 l}{8}}\right) = \mathbb{E}\left(\mathbb{I}\left\{N_t = l\right\} \cdot e^{xZ_t - \frac{x^2(M-m)^2 N_t}{8}}\right)$$

$$\le \mathbb{E}\left(e^{xZ_t - \frac{x^2(M-m)^2 N_t}{8}}\right)$$

$$= \mathbb{E}(M_t)$$

$$\le \mathbb{E}(M_0) = 1.$$

This yields

$$\mathbb{P}\left(Z_t \ge \epsilon \text{ and } N_t = l\right) \le e^{-\epsilon x + \frac{x^2(M-m)^2 l}{8}}.$$

In particular, for $\epsilon = (M-m)\sqrt{\frac{l \cdot \log(1/\delta)}{2}}$ and $x = \frac{4\epsilon}{l(M-m)^2}$,

$$\mathbb{P}\left(Z_t \ge (M-m)\sqrt{\frac{l \cdot \log(1/\delta)}{2}} \text{ and } N_t = l\right) \le \delta.$$

This proves the first part of the Lemma. Summing over the values of $l$ from $1$ to $t$, we find that

$$\mathbb{P}\left(Z_t \ge (M-m)\sqrt{\frac{N_t \log(1/\delta)}{2}} \text{ and } N_t \ge 1\right) \le t\delta.$$

Similar arguments can be used to prove that

$$\mathbb{P}\left(-Z_t \ge (M-m)\sqrt{\frac{N_t \log(1/\delta)}{2}} \text{ and } N_t \ge 1\right) \le t\delta.$$

In order to conclude the proof we observe that $Z_t = \hat{\mu}_t N_t$, we normalize by $N_t$, and observe that adding the case $N_t = 0$ can only increase the probability. $\qquad\square$

### E.12 PROOF OF LEMMA 14

**Lemma 14.** *On the event $\mathcal{E}^{\mathrm{ETC}}$, for all $(k, k') \in \mathcal{A}_t$, and $p = x_t^\top \widehat{\theta}_t^s + k\epsilon$, we have*

$$\left| \mathrm{EGFT}(x_t, p) - \left( \widehat{I}^{k'} \widehat{F}^k + \widehat{J}^k \widehat{D}^{k'} \right) \right| \leq 6PL\epsilon + 3\epsilon + 2P \left( |\widehat{\Delta}F| + |\widehat{\Delta}D| \right) + |\widehat{\Delta}I| + |\widehat{\Delta}J|$$

*where we define $\widehat{\Delta}I = I(k'\epsilon) - \widehat{I}^{k'}$, $\widehat{\Delta}J = J(k\epsilon) - \widehat{J}^k$, $\widehat{\Delta}F = F(k\epsilon) - \widehat{F}^k$, $\widehat{\Delta}D = D(k'\epsilon) - \widehat{D}^{k'}$.*

*Proof.* By Lemma 1, for any price $p = x_t^\top \widehat{\theta}_t^s + k\epsilon$ and $k'$ such that $(k, k') \in \mathcal{A}_t$, and $\delta^s = p - x_t^\top \theta^s$, $\delta^b = p - x_t^\top \theta^b$, we have that

$$\mathrm{EGFT}(x_t, p) = I(\delta^b)F^s(\delta^s) + J(\delta^s)D^b(\delta^b).$$

Then,

$$\mathrm{EGFT}(x_t, p) = \left( I(k'\epsilon) + I(\delta^b) - I(k'\epsilon) \right) \left( F^s(k\epsilon) + F^s(\delta^s) - F^s(k\epsilon) \right) +$$
$$\left( J(k\epsilon) + J(\delta^s) - J(k\epsilon) \right) \left( D^b(k'\epsilon) + D^b(\delta^b) - D^b(k'\epsilon) \right).$$

Moreover, by letting $\Delta I = I(\delta^b) - I(k'\epsilon)$, $\Delta F = F^s(\delta^s) - F^s(k\epsilon)$, $\Delta J = J(\delta^s) - J(k\epsilon)$ and $\Delta D = D^b(\delta^b) - D^b(k'\epsilon)$, this yields

$$\mathrm{EGFT}(x_t, p) = I(k'\epsilon)F^s(k\epsilon) + J(k\epsilon)D^b(k'\epsilon) + I(\delta^b)\Delta F +$$
$$F^s(k\epsilon)\Delta I + J(\delta^s)\Delta D + D^b(k'\epsilon)\Delta J.$$

Since $I$ and $J$ are bounded by $2P$, and $F$ and $D$ are bounded by $1$, this implies that

$$\left| \mathrm{EGFT}(x_t, p) - \left( I(k'\epsilon)F^s(k\epsilon) + J(k\epsilon)D^b(k'\epsilon) \right) \right| \leq 2P|\Delta F| + |\Delta I| + 2P|\Delta D| + |\Delta J|.$$

Now, let us introduce $e^s = \delta^s - k\epsilon$, and $e^b = \delta^b - k'\epsilon$. Then, we have

$$e^s = \left( x_t^\top \widehat{\theta}_t^s + k\epsilon \right) - \left( k\epsilon + x_t^\top \theta^s \right)$$
$$= x_t^\top \left( \widehat{\theta}_t^s - \theta^s \right).$$

On event $\mathcal{E}^{\mathrm{ETC}}$ we have $|e^s| \leq \epsilon$. Since $F$ is $L$-Lipschitz continuous, and $\Delta F = F^s(k\epsilon + e^s) - F^s(k\epsilon)$, this implies $|\Delta F| \leq L\epsilon$. Similarly, $J$ is 1-Lipschitz continuous, and $\Delta J = J(k\epsilon + e^s) - J(k\epsilon)$, so $|\Delta J| \leq \epsilon$. Similarly,

$$e^b = \left( x_t^\top \widehat{\theta}_t^s + k\epsilon \right) - \left( x_t^\top \theta^b + k'\epsilon \right)$$
$$= \left( x_t^\top (\widehat{\theta}_t^s - \widehat{\theta}_t^b) + k\epsilon \right) - k'\epsilon + x_t^\top (\widehat{\theta}_t^b - \theta^b).$$

By definition of $\mathcal{A}_t$, under the event $\mathcal{E}^{\mathrm{ETC}}$ we have $|e^b| \leq 2\epsilon$. Since $D$ is $L$-Lipschitz continuous and $\Delta D = D^b(k'\epsilon + e^b) - D^b(k'\epsilon)$, $|\Delta D| \leq 2L\epsilon$. Similarly, $I$ is 1-Lipschitz continuous, $\Delta I = I(k'\epsilon + e^b) - I(k'\epsilon)$, so this implies that $|\Delta I| \leq 2\epsilon$. Putting everything together, we find that on the event $\mathcal{E}^{\mathrm{ETC}}$,

$$\left| \mathrm{EGFT}(x_t, p) - \left( I(k'\epsilon)F^s(k\epsilon) + J(k\epsilon)D^b(k'\epsilon) \right) \right| \leq 6PL\epsilon + 3\epsilon.$$

Similarly, denoting $\widehat{\Delta}I = I(k'\epsilon) - \widehat{I}^{k'}$, $\widehat{\Delta}D = D(k'\epsilon) - \widehat{D}^{k'}$, $\widehat{\Delta}J = J(k\epsilon) - \widehat{J}^k$ and $\widehat{\Delta}F = F(k\epsilon) - \widehat{F}^k$, we have

$$\left| I(k'\epsilon)F^s(k\epsilon) + J(k\epsilon)D^b(k'\epsilon) - \left( \widehat{I}^{k'} \widehat{F}^k + \widehat{J}^k \widehat{D}^{k'} \right) \right| \leq 2P|\widehat{\Delta}F| + |\widehat{\Delta}I| + 2P|\widehat{\Delta}D| + |\widehat{\Delta}J|.$$

This concludes the proof. $\square$

## F  MOTIVATING EXAMPLE

Lemma 1 emphasizes that the expected gain from trade at a given price $p$ depends on the quantities $\delta^s = p - x_t^\top \theta^s$ and $\delta^b = p - x_t^\top \theta^b$. Remember that the difference in average valuations $\Delta$ is given by $\Delta = x_t^\top \theta^b - x_t^\top \theta^s$, and with this notation, $\delta^b = \delta^s - \Delta$. Therefore, the expected gain from trade can be rewritten as a function of the pair $(\delta^s, \Delta)$. As an immediate consequence, we see that the optimal increment $\delta^s$ only depends on the difference in average valuations $\Delta$: if $p = x^\top \theta^s + \delta^s$ maximizes EGFT$(x, p)$, and if $x'$ is such that $x'^\top \theta^b - x'^\top \theta^s = x^\top \theta^b - x^\top \theta^s$, then $p' = x'^\top \theta^s + \delta^s$ maximizes EGFT$(x', p')$.

On the other hand, the following example shows that there no explicit dependence of the optimal price increment $\delta^s$ on the difference in average valuations $\Delta$. In words, when $\Delta$ is small, we might prefer to choose an increment $\delta^s$ that leads to trade happening with lower probabilities but corresponds to higher rewards. By contrast, as $\Delta$ increases, similar values of the increment $\delta^s$ will correspond to higher gains if the trade happens. Then, we might choose to post prices corresponding to a smaller increment $\delta^s$ to increase the probability that the trade happens.

This reasoning demonstrates that knowing that an increment $\delta^s$ is optimal for a difference in average valuations $\Delta$ does not allow us to determine the optimal increment $\delta'$ corresponding to a different value $\Delta'$ of the difference in average valuations. This implies that to precisely identify the optimal price increment $\delta^s$ for all differences in average valuations $\Delta$, it may be necessary to have accurate estimates of the functions $F^s$ and $I$ for a broad range of values of $\delta^s$. Similar arguments can be employed to argue that precise estimates of the value of $D^b$ and $J$ for a wide range of values of $\delta^b = \Delta - \delta^s$ might also be necessary.

To illustrate this phenomenon, we construct an example where different levels of $\Delta$ lead to entirely different choices of the optimal price increment $\delta^s$. Specifically, we consider a scenario where, for certain values of $(s_1, s_2, s_3) \in \mathbb{R}^3$, $(b_1, b_2, b_3) \in \mathbb{R}^3$, $(\alpha_1, \alpha_2, \alpha_3)$ in the simplex, and $\theta > 0$ (to be defined later), the density $f^s$ (resp. $f^b$) of the seller's noise $\xi^s$ (resp. buyer's noise $\xi^b$) is given by:

$$f^s(\delta) = \frac{\mathbb{I}\{x \in [s_1, s_1 + \theta] \cup [s_2, s_2 + \theta] \cup [s_3, s_3 + \theta]\}}{3\theta}$$
$$f^b(\delta) = \frac{\alpha_1 \mathbb{I}\{x \in [b_1, b_1 + \theta]\} + \alpha_2 \mathbb{I}\{x \in [b_2, b_2 + \theta]\} + \alpha_3 \mathbb{I}\{x \in [b_3, b_3 + \theta]\}}{\theta}.$$

The setting in illustrated in figure 1.

We assume that $(s_1, s_2, s_3)$ and $(b_1, b_2, b_3)$ verify $s_1 + \theta < b_1, b_1 + \theta < s_2, s_2 + \theta < b_2, b_2 + \theta < s_3$, and $s_3 + \theta < b3$. Then, it is straightforward to see that while $\Delta > 0$, and $\Delta$ is small enough so that $b_1 + \theta + \Delta < s_2$ and $b_2 + \theta + \Delta < s_3$, the optimal increment belongs to the set $\{\delta_1, \delta_2, \delta_3\}$, where $\delta_1$, $\delta_2$, and $\delta_3$ belong respectively to $[s_1 + \theta, b_1]$, $[s_2 + \theta, b_2]$, and $[s_3 + \theta, b_3]$.

We also assume that $\alpha_1$ is much larger than $\alpha_2$, which in turn is much larger than $\alpha_3$. Then, the increment $\delta_1$ is such that the trade happens with the highest probability: indeed, if $\delta^s = \delta^b > b_1$, the buyer rejects the trade with a high probability. For the same reasons, the probability of a trade happening at an increment $\delta_2$ is much lower, and the probability of a trade happening at increment $\delta_3$ is the lowest.

Finally, we assume that $b_3 - s_3$ is much larger than $b_2 - s_2$, which in turn is much larger than $b_1 - s_1$. Then, the gain from any trade happening at increment $\delta_1$ is small compared to the gain from trades happening at increment $\delta_2$, which in turn is small compared to the gain from a trade happening at increment $\delta_3$.

For some well-chosen values of the parameters specified below, when the average gain from trade of the seller and the buyer are both zero ($x_t^\top \theta^s = x_t^\top \theta^b = 0$), the most profitable increment is $\delta_3$: the probability of the trade happening is less likely, but when it occurs, it is more profitable.

Now, to obtain the probability of a price $p = x^\top \theta^s + \delta^s$ being accepted by the buyer, it is sufficient to translate the cumulative distribution function (c.d.f.) of $\xi^b$ (represented in Figure 1) by the quantity $\Delta = x^\top (\theta^b - \theta^s)$. In particular, when $\Delta > 0$ is small enough so that $b_1 + \theta + \Delta < s_2$ and $b_2 + \theta + \Delta < s_3$, we observe that, on the one hand, the probability that the trade happens for the increments $\delta^s \in \delta_1, \delta_2, \delta_3$ remains unchanged; however, the corresponding gains if the trade occurs all increase with $\Delta$. For some well-chosen values of the parameters, if $\Delta$ is positive but sufficiently

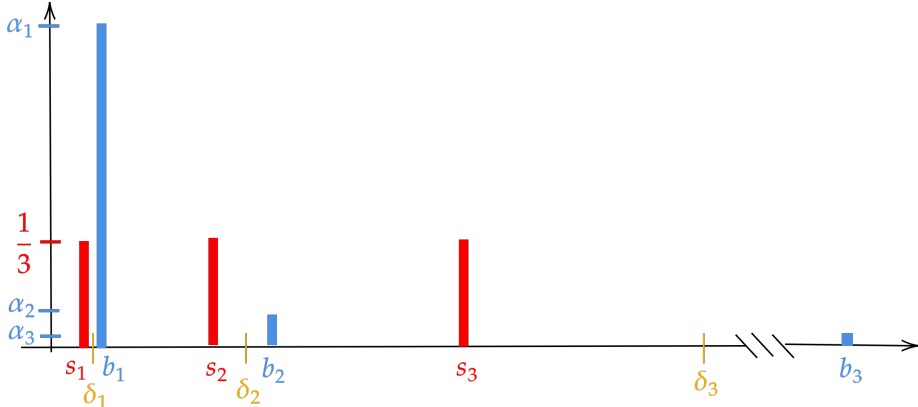

Figure 1: Illustration for the c.d.f. $f^s$ and $f^b$ corresponding to the example described in Section F. The c.d.f. $f^s$ is represented in red, the c.d.f. $f^b$ is represented in blue.

small, the expected gain from trade is maximized by $\delta^s = \delta^2$. When $\Delta$ is large, the expected gain from trade is maximized by choosing $\delta^s = \delta^1$: in other words, this choice of increment ensures that the sale happens with the highest probability, and each sale leads to a reward of at least $\Delta$. In Figure 2, we plot the expected gain from trade for different values of $\Delta$ as a function of $\delta^s$. The values chosen for the parameters are as follows: $(s_1, s_2, s_3) = (0, 2, 6)$, $(b_1, b_2, b_3) = (0.01, 3, 20)$, $(\alpha_1, \alpha_2, \alpha_3) = (0.85, 0.11, 0.04)$, and $\theta = 0.001$. With these values, for $\Delta = 0$, the optimal increment is $\delta^3 = 10$; for $\Delta = 1$, it is $\delta^2 = 2.5$; and for $\Delta = 1.5$, it is $\delta^3 = 0.01$.

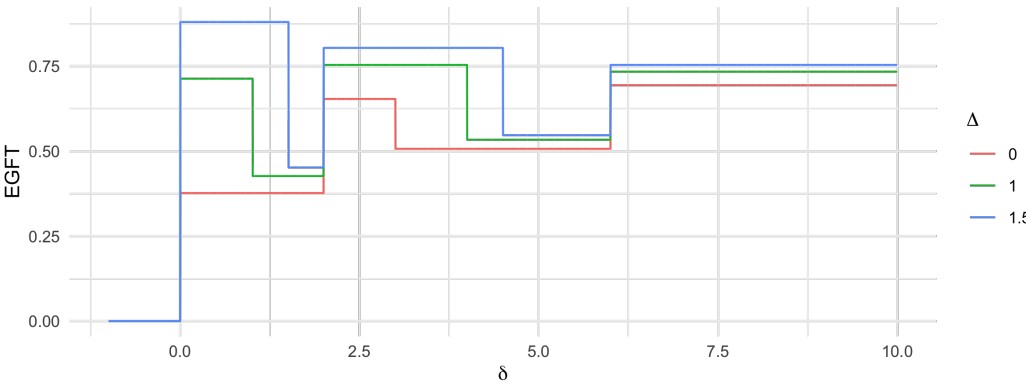

Figure 2: Expected gain from trade for different values of increment $\delta^s$ and difference in average valuations $\Delta$.

## G   PROOFS OF SECTION 5

**Lemma 2.** *For each round $t \in \mathcal{T}^B$ such that $b_t \geq s_t$, it holds:* $\mathbb{E}[\text{PROFIT}_t(p_t, q_t)] \geq \frac{(b_t - s_t)^2}{8P \log T} - \frac{2}{T}$, *where the expectation is with respect to the choice of $(p_t, q_t)$.*

*Proof.* To simplify notation, and since we focus on a single round $t \in \mathcal{T}^B$, we omit the explicit dependence on $t$ from $p_t$, $q_t$, $i_t$, $s_t$, and $b_t$. We consider two cases. First, if $b - s \leq 2/T$, the inequality is immediately satisfied since the lhs is non-negative while the rhs is $\leq 0$. Second, if $b - s > 2/T$,

let $\mathcal{E}$ be the event in which $p \in [s, (s + b)/2]$. Since $p \sim \mathcal{U}([0, P])$ we have $\mathbb{P}(\mathcal{E}) = (b - s)/2P$. Moreover, we have

$$\mathbb{P}\left(i = \left\lfloor \log \frac{1}{b - p} \right\rfloor \mid A\right) = \frac{1}{\log T},$$

which is well defined since

$$p \leq \frac{s + b}{2} \leq b - \frac{1}{T}.$$

Let $\mathcal{E}'$ denote the event in which $i = \lfloor \log(1/(b - p)) \rfloor$. Under $\mathcal{E}$ and $\mathcal{E}'$ we get

$$q = p + 2^{-i} = p + 2^{-\lfloor \log(1/(b-p)) \rfloor} \geq p + 2^{-(\log(1/(b-p))+1)} = p + \frac{b - p}{2} = \frac{p + b}{2}.$$

Therefore, when $b \geq s + 2/T$,

$$\mathbb{E}[\textsc{Profit}(p, q)] = (q - p) \frac{b - s}{2P \log T}$$

$$\geq \left(\frac{p + b}{2} - p\right) \frac{b - s}{2P \log T}$$

$$\geq \frac{(b - s)^2}{8P \log T}.$$

This concludes the proof. $\qquad\square$

**Lemma 3.** *For $\tau \geq \log T$, it holds with probability at least $1 - 1/T$ that*

$$\sum_{t \in [\![\tau]\!]} \textsc{Profit}_t(p_t, q_t) \geq \frac{\alpha}{8P \log(T)} \sum_{t \in [\![\tau]\!]} [b_t - s_t]^+ - \sqrt{4P^2 \log(T) \sum_{t \in [\![\tau]\!]} [b_t - s_t]^+} - 2.$$

*Proof.* First, we observe that given the sequence $(b_t, s_t)_{t \in [\![T]\!]}$, and a time step $\tau$, by Azuma-Hoeffding inequality we have that, with probability at least $1 - \delta$,

$$\sum_{t \in [\![\tau]\!]} \textsc{Profit}_t(p_t, q_t) \geq \mathbb{E}[\textsc{Profit}_t(p, q)] - \sqrt{2 \log(1/\delta) \sum_{t \in [\![\tau]\!]} ([b_t - s_t]^+)^2}.$$

Then, by applying a union bound, the above inequality holds simultaneously for all possible $\tau$ with probability at least $1 - \delta T$. Then, by setting $\delta = T^2$, with probability at least $1 - 1/T$, it holds:

$$\sum_{t \in [\![\tau]\!]} \textsc{Profit}_t(p_t, q_t)$$

$$\geq \sum_{t \in [\![\tau]\!]} \mathbb{E}[\textsc{Profit}_t(p, q)] - \sqrt{4 \log(T) \sum_{t \in [\![\tau]\!]} ([b_t - s_t]^+)^2}$$

$$\geq \sum_{t \in [\![\tau]\!]} \left(\frac{([b_t - s_t]^+)^2}{8P \log T} - \frac{2}{T}\right) - \sqrt{4 \log(1/\delta) \sum_{t \in [\![\tau]\!]} ([b_t - s_t]^+)^2} \qquad \text{(by Lemma 2)}$$

$$\geq \sum_{t \in [\![\tau]\!]} \frac{([b_t - s_t]^+)^2}{8P \log T} - \sqrt{4 \log(T) \sum_{t \in [\![\tau]\!]} ([b_t - s_t]^+)^2} - 2$$

$$= \tau \sum_{t \in [\![\tau]\!]} \frac{1}{\tau} \frac{([b_t - s_t]^+)^2}{8P \log T} - \sqrt{4 \log(T) \sum_{t \in [\![\tau]\!]} ([b_t - s_t]^+)^2} - 2$$

$$\geq \frac{\tau}{8P \log T} \left(\sum_{t \in [\![\tau]\!]} \frac{[b_t - s_t]^+}{\tau}\right)^2 - \sqrt{4 \log(T) \sum_{t \in [\![\tau]\!]} ([b_t - s_t]^+)^2} - 2 \qquad \text{(by Jensen's Inequality)}$$

$$\geq \frac{\alpha}{8P \log(T)} \sum_{t \in [\![\tau]\!]} [b_t - s_t]^+ - \sqrt{4 \log(T) \sum_{t \in [\![\tau]\!]} ([b_t - s_t]^+)^2} - 2 \qquad \text{(by Definition 1)}$$

$$\geq \frac{\alpha}{8P \log(T)} \sum_{t \in [\![\tau]\!]} [b_t - s_t]^+ - \sqrt{4P^2 \log(T) \sum_{t \in [\![\tau]\!]} [b_t - s_t]^+} - 2.$$

This concludes the proof. □

**Theorem 3.** *Given the two-bit algorithm $\mathcal{A}$, the corresponding one-bit learning algorithm satisfies global budget balance and, with probability at least $1 - 1/T$, has regret*

$$R_T^{(1\,\text{bit})} \leq O\left(\alpha^{-3} T^{\text{E}} \log^4 T\right) + R_T^{(2)}.$$

*Proof.* The one-bit algorithm is global budget balanced by construction (see choice of $\mathcal{B}$).

Then, we condition the high probability regret bound on the following events:

- With probability $1 - 1/T$ the two-bit EOC algorithm guarantees a number of exploration rounds smaller than $|\mathcal{T}^{\text{E}}|$ and regret at most $R_T^{(2)}$;

- With probability $1 - 1/T$, it holds the inequality in Lemma 3

- By Azuma-Hoeffding, with probability at least $1 - 1/T$ it holds

$$\sum_{t=1}^{\tau} \max_p \text{EGFT}(x_t, p) \leq \sum_{t=1}^{\tau} [b_t - s_t]^+ + \sqrt{16 P^2 \tau \log(T)}.$$

Then, the regret can be bounded as follows

$$
\begin{aligned}
R_T &= \sum_{t=1}^{T} \max_p \text{EGFT}(x_t, p) - \sum_{t=1}^{T} \text{EGFT}(x_t, p_t) \\
&\leq \sum_{t=1}^{\tau} \max_p \text{EGFT}(x_t, p) + \sum_{t=\tau+1}^{T} \left( \max_p \text{EGFT}(x_t, p) - \text{EGFT}(x_t, p_t) \right) \\
&\leq \sum_{t=1}^{\tau} \max_p \text{EGFT}(x_t, p) + 2 T^{\text{E}} + R_T^{(2)} \\
&\leq \sum_{t=1}^{\tau} [b_t - s_t]^+ + \sqrt{16 P^2 \tau \log(T)} + 2 T^{\text{E}} + R_T^{(2)} \\
&\leq \sum_{t=1}^{\tau} [b_t - s_t]^+ + \alpha \tau + 2 T^{\text{E}} + R_T^{(2)} \\
&\leq 2 \sum_{t=1}^{\tau} [b_t - s_t]^+ + 2 T^{\text{E}} + R_T^{(2)},
\end{aligned}
$$

where in the second-to-last inequality we use that, since $\tau \geq \mathcal{B}/2P$, it holds that $\tau \geq 16 P^2 \alpha^{-2} \log T$ (by Equation (5)). The last inequality is by definition of $\alpha$ on the interval $[\![\tau]\!]$.

Then, by Lemma 3 and since $\sum_{t=1}^{\tau} [b_t - s_t]^+ \geq \sum_{t=1}^{\tau} \text{PROFIT}_t(p_t, q_t) \geq 2048 P^4 \alpha^{-2} \log^3 T$, we have

$$
\begin{aligned}
\sum_{t\in[\![\tau]\!]} \text{PROFIT}_t(p_t, q_t) &\geq \frac{\alpha}{8 P \log(T)} \sum_{t\in[\![\tau]\!]} [b_t - s_t]^+ - \sqrt{4 P^2 \log(T) \sum_{t\in[\![\tau]\!]} [b_t - s_t]^+} - 2 \\
&\geq \frac{\alpha}{8 P \log(T)} \sum_{t\in[\![\tau]\!]} [b_t - s_t]^+ - \sqrt{8 P^2 \log(T) \sum_{t\in[\![\tau]\!]} [b_t - s_t]^+} \\
&\geq \frac{\alpha}{16 P \log(T)} \sum_{t\in[\![\tau]\!]} [b_t - s_t]^+.
\end{aligned}
$$

Then, since $\mathcal{B} \le 2048 P^4 \alpha^{-2} T^{\mathrm{E}} \log^3 T$ and $\sum_{t=1}^{\tau} \mathrm{PROFIT}_t(p_t, q_t) \le \mathcal{B} + 2P$, by substituting in the expression above we obtain the desired bound on $\sum_{t \in [\![\tau]\!]} [b_t - s_t]^+$. In particular, we have

$$
\begin{aligned}
R_T &\le 2 \sum_{t=1}^{\tau} [b_t - s_t]^+ + 2|\mathcal{T}^{\mathrm{E}}| + R_T^{(2)} \\
&\le \frac{32 P \log T}{\alpha} \sum_{t \in [\![\tau]\!]} \mathrm{PROFIT}_t(p_t, q_t) + 2 T^{\mathrm{E}} + R_T^{(2)} \\
&\le M \frac{P^5 \log^4 T}{\alpha^3} T^{\mathrm{E}} + 2 T^{\mathrm{E}} + R_T^{(2)},
\end{aligned}
$$

with $M$ is a numeric constant independent on the problem instance. $\qquad \square$

## H  ALTERNATIVE ALGORITHM FOR NOISY VALUATIONS WITH TWO-BIT FEEDBACK

The regret of Algorithm 3 is primarily driven by the estimation of the c.d.f. $F^s$ and demand function $D^b$ uniformly over a grid of increments. We now present an alternative sub-optimal approach. Specifically, to leverage the fact that the optimal increment $\delta^s$ only depends on the difference in average valuations $\Delta_t$, we could first execute the sub-routines EST-PAR and EST-INT, yielding estimates of $x_t^\top \theta^s$, $x_t^\top \theta^b$, $I$ and $J$ up to precision $\epsilon$ using $\widetilde{O}(\epsilon^{-2})$ samples. Then, we could round the value of $\Delta_t$ on a grid of size $\epsilon^{-1}$, and run independent Scouting Bandit algorithms (as described in Cesa-Bianchi et al. (2024)) for each of the $\epsilon^{-1}$ rounded values.

The grid size implies a discretization error of order $O(\epsilon)$. The highest regret occurs when each of the $\epsilon^{-1}$ independent Scouting Bandit algorithms runs for the same number of rounds $T_\epsilon = T\epsilon$. Each algorithm incurs a regret $\widetilde{O}(T_\epsilon^{2/3})$, so their combined regret is $\widetilde{O}(T_\epsilon^{2/3} \cdot \epsilon^{-1}) = \widetilde{O}(T^{2/3} \epsilon^{-1/3})$. For $\epsilon = T^{-1/4}$, this strategy also results in a regret of order $\widetilde{O}(T^{3/4})$.

