# OpenReview forum: "Feature-Based Online Bilateral Trade"
_ICLR.cc/2025/Conference — ICLR 2025 Poster_

### Official Review · Reviewer_HHmH · 2024-11-01

**Soundness:** 4
**Presentation:** 3
**Contribution:** 3
**Rating:** 8
**Confidence:** 3

**Summary:**

The authors study the feature based online bilateral trade problem. Here,  the buyer's valuations are given by a linear function. The seller's valuation is similar. In each round, the buyer and seller see a new context that along with their private parameter vector determines the parameterized part of the reward. The noisy version of the problem adds a i.i.d. random variable (not necessarily zero mean) with the parameterized reward.  The authors study the problem in both 2-bit and 1-bit feedback under strong and global budget balance constraints, respectively. For the deterministic version they adopt existing  EllipsoidPricing policy and show a log(T) regret bound. For the noisy, version they propose an Explore-or-Commit algorithm that achieves a $O(T^{3/4})$ regret, which is further improved to $O(T^{2/3})$ (which matches the lower bound). Some tradeoff between budget and regret is established for 1-bit feedback case.

**Strengths:**

- This work initiates the feature based online bilateral trade problem
- With budget balance and two-bit feedback they establish a tight $O(T^{2/3})$ regret bound with by combining Scouting and Explore-or-Commit strategies.
- They extend the results to the one-bit feedback setup while maintaining the regret guarantees under the budget balance constraint.

**Weaknesses:**

- See questions for more discussions around improving the paper, and my own curiosity.

**Questions:**

Two bit feedback:
- Do we gain anything by removing the budget balance constraint in the two-bit feedback case (simplifying the algorithm maybe)?
- Can the authors provide more intuition of how combining the ETC, and Scouting strategy works? (maybe along the line -> with $O(T^\beta)$  exploration regret we reduce the 'Range of Delta_t' = $O(T^{-\alpha})$ and then the Scouting results in $O(T^{2/3})$ regret)
- As we do not rely on the exact reward feedback, will approximately linear reward functions work?


One bit feedback:
- In the one-bit feedback case, is the knowledge of $\alpha$ essential?
- Can the authors discuss if the $\alpha$ dependency a side effect of selecting the specific strategy of collecting the profit in the first phase? Can we improve/remove such dependency by adaptively collecting the budget or by leveraging improved exploration strategy?

---

> ### Author Response · Authors · 2024-11-21
>
> We thank the reviewer for the positive feedback about the paper.
>
> **1)** Posting two different prices without a strict budget balance in the two-bit feedback can indeed make exploration more efficient. This approach allows us to simultaneously estimate the quantities of interest (expected valuations and the CDF of the noise) for both the buyer and the seller. In contrast, posting the same price for both forces us to separate the estimation phases for the buyer's and seller's parameters. However, note that this change does not affect the order of the regret, as it only reduces the length of the exploration phase by half. To achieve the lower regret of order $\sqrt{T}$, full observation of the valuations is required, as discussed by Cesa-Bianchi et al. in the context of non-contextual bilateral trade. In summary, we would not achieve any improvement, at least in terms of the order of regret.
>
> **2)** Intuitively, scouting bandits (combined with information pooling) is a tool to avoid performing a pure exploration phase (as in ETC algorithms). We refer the reader to the sketch of proof following Theorem 3 for intuitions on the regret of the Scouting phase and to the sketch of proof following Theorem 2 for intuitions on the regret of the estimation phases of the parameters $\theta^s$ and $\theta^b$, and of the functions $I$ and $J$.
>
> **3)** Approximately linear reward functions should also work, as long as the deviation from the linear model is not too large. Just as we can handle noise-induced corruption, we can also accommodate small corruptions in the model. However, if the deviations from the linear model are significant, the analysis is likely to fail, and we leave studying this new setup as an interesting direction for future work.
>
> **4)** We distinguish between the existence of an $\alpha > 0$ and knowledge of such an $\alpha$. The former is necessary, as without it, the learner would be unable to build up a budget, making the tradeoff between budget and information gain impossible. However, in principle, knowledge of $\alpha$ is not required. Indeed, it should be possible to dynamically estimate $\alpha$ by dividing the time horizon into windows and adjusting the estimate based on the observations within each window. We leave this an interesting direction for future research.
>
> **5)** The dependency on $\alpha$ of the regret bound is unavoidable because, intuitively, when $\alpha$ approaches zero gathering the required budget becomes impossible. Moreover, we conjecture that the dependency on $\alpha$ in Lemma 3 is tight. Hence, intuitively, $(b_t-s_t)^2$ is too small to collect enough budget when $\alpha$ is small.

---

> > ### Comment · Reviewer_HHmH · 2024-11-21
> > **Response**
> >
> > I thank the authors for their insightful response. I encourage the authors to add the discussions about future directions in the paper. The dependence of $\alpha$ can be improved by adding the discussions above in the paper. I will maintain my score.

---

### Official Review · Reviewer_CL94 · 2024-11-02

**Soundness:** 2
**Presentation:** 3
**Contribution:** 2
**Rating:** 6
**Confidence:** 3

**Summary:**

This paper studies the bilateral trade model which involves the challenge of enabling transactions between a seller and a buyer who both hold private valuations for the item. This paper considers specifically the online scenario, where at each step there is a fresh seller and buyer entering the system and the pricing decisions for both parties must be made immediately without prior knowledge of their valuations. The paper further restricts to the contextual setting where the private valuations for the seller and buyer are linearly featured by a context. A two-bid feedback is considered where for both the seller and the buyer, it can be observed whether the posed price has exceeded its value or not. By further assuming a strong budget balance between the buyer and the seller, the paper is able to derive a $O(\log T)$ regret. Without the budget balance conditions, the paper achieves a $O(T^{2/3})$ regret, which is minimax optimal. The paper further discusses the one-bit feedback and shows the potential to obtain a sub-linear regret.

**Strengths:**

1. The paper derives strong $O(\log T)$ regret, though under stronger conditions.

2. The paper derives $O(T^{2/3})$ regret upper bound for their algorithm and shows that there exists a matching lower bound.

**Weaknesses:**

1. The main results of the paper rely on the two-bid feedback setting, where both the seller and the buyer reveal to the decision maker whether they want to sell the product or buy the product. This is quite a strong condition, and the paper would benefit from a more detailed discussion on whether this condition happens or not in reality.

2. Though the theoretical guarantee is provided, there are not numerical experiments in the paper showing the empirical performances. Also, the computation complexity of the proposed algorithms has not been discussed.

3. The algorithmic idea and the proof technique mainly build upon the previous work Cohen et al. (2020) and it has not been discussed which part of the proof is novel.

4. The $O(\log T)$ regret depends on some strong conditions that are hard to justify in practice.

5. The paper is overall theoretical and it is not clear how to apply their algorithm in practice.

**Questions:**

1. How do you check whether the strong budget balance condition holds or not?

2. Could you please conduct numerical experiments to show the real performance? Also, what is the computation complexity of your algorithms.

---

> ### Author Response · Authors · 2024-11-21
>
> We thank the reviewer for their feedback about the paper. We will start by addressing the weaknesses identified, and then we will address the reviewer’s questions.
>
> **W1)** The two-bit feedback condition is a relatively mild requirement and is a reasonable assumption for most practical applications. Specifically, the learner only needs to observe whether each agent accepted or rejected the trade, which is typical in platforms that use posted price mechanisms, where both the buyer and seller independently decide to accept or decline the posted price. Indeed, this type of feedback is commonly referred to as "realistic feedback" in the literature (see, e.g., Cesa-Bianchi et al. (2021)).
>
> **W2)** See question.
>
> **W3)** and **W4)** The results for the noiseless setting are included only for completeness and are clearly marked as a “Warm up” (see Section 3), since their derivations are, indeed,  rather straight forward from Cohen et al. (2020). But apart from these warm-up results, none of our main contributions for the noisy setting draws upon the results of  Cohen et al. (2020).
>
> **W5)** We agree that the contributions of this paper are primarily theoretical, as we analyze the problem and characterize its difficulty in terms of regret under various assumptions. We consider this a necessary step to gain a deeper understanding of the problem before exploring practical applications. Additionally, we would like to highlight that such theoretical contributions are encouraged by the ICLR guidelines.
>
> **Q1)** We’re not entirely sure we fully understand the reviewer’s question, so please let us know if we are misinterpreting it. Budget balance is something the learner must enforce when determining the prices to be posted. Specifically, if the prices posted to the seller and the buyer are the same, then strong budget balance is satisfied.
>
>
> **Q2)** We interpret "computational complexity" as referring to the "per-iteration running time." If that is correct, the per-iteration running time is polynomial in the size of the problem and independent of $T$. This aligns with the standard requirement for online learning and bandit algorithms.
>
> We did not include numerical experiments for two main reasons: 1) the primary goal of the paper is to provide a theoretical characterization of the problem, which we consider a crucial step before exploring practical applications; 2) our algorithms are the first to be proposed and analyzed for this specific setup, meaning we lack a meaningful baseline for conducting experiments. However, our algorithms can serve as a baseline for future experimental evaluations of other approaches.

---

> > ### Comment · Reviewer_CL94 · 2024-11-27
> >
> > Thank you for the detailed response. I will maintain my score.

---

### Official Review · Reviewer_Pddv · 2024-11-10

**Soundness:** 3
**Presentation:** 3
**Contribution:** 3
**Rating:** 6
**Confidence:** 4

**Summary:**

In this paper, the authors investigate online contextual bilateral trade problem, where the valuations of two traders are modeled by different (unknown) linear functions. The authors focus on two different feedback models: (1) two-bit feedback model, where the learner can observe the binary feedback of both traders (2) one-bit feedback model, where the learner can only learns the binary information of whether the trade happens or not. For (1), the authors propose an online learning algorithm to set trading price at each round (that satisfies the strong budget balance constraint), which achieves $O(T^{2/3})$ regret. The authors also show a matching lower bound. For (2), the authors provide a reduction from one-bit feedback model to two-bit feedback model by sacrificing per-round budget balance to global budget balance and show the algorithm used in (1) can be applied to (2) and still achieves sublinear regret guarantee.

**Strengths:**

The paper is well-written and analyzes a very interesting theoretical problem. The authors did a good job to describe the problem and how the algorithm handles the challenges.

The theoretical guarantee of the paper is sound. The authors provide a complete story for the setting with two-bit feedback model. In addition, the reduction from one-bit to two-bit by sacrificing budget balance constraint is very interesting and elegant.

**Weaknesses:**

There is no matching lower bound for the one-bit feedback setting. I also have some questions regarding this setting.

**Questions:**

1. Can you elaborate a bit more about the comparison with the related work, "A contextual online learning
theory of brokerage. arXiv preprint arXiv:2407.01566, 2024"? The setting is very similar, however, it seems the valuations of two traders in their paper share the same expected value.

2. For the one-bit feedback model, if we want to maintain per-round budget balance, is it still learnable?

---

> ### Author Response · Authors · 2024-11-21
>
> We thank the reviewer for the positive feedback about the paper.
>
> **1)** The online brokerage model has several fundamental differences from our model. We will discuss  this in the final version of the paper. However, we emphasize that the two problems are related only at a high level and present significantly different challenges.
>
> In online brokerage, the agents have the same expected valuation for the good, and their roles as buyer or seller are determined based on their valuations of the good relative to each other. Specifically, at each round, agents $q$ and $q'$ have valuations $v$ and $v'$. If $v < v'$, $q$ acts as the seller and $q'$ as the buyer; if $v > v'$, the roles are reversed. In contrast, the bilateral trade problem assumes that buyer and seller roles are predetermined, typically reflecting a situation where one party owns a good that the other seeks to purchase.
>
> These two models not only represent different situations but also differ significantly in their difficulty. In Bachoc et al., when both agents share the same expected valuation, the optimal price is simply this shared value, making the problem parametric and requiring only the learning of this expected value. Moreover, the regret is quadratic in the price error, allowing for regret rates that scale as $\sqrt{T}$ up to logarithmic factors.
>
> In the bilateral trade problem, the buyer and seller have distinct average valuations, and the optimal price depends on the noise distributions affecting each party's valuation. Here, estimating the linear expected valuations (using techniques similar to those in Bachoc et al.) is the ``easy part’’ of the problem; the more challenging, non-parametric task is determining the optimal price. This requires estimating the cumulative distribution function of the noise. Pure exploration, as done in our Explore or Commit algorithm, yields a suboptimal regret rate. Our Scouting Bandit with Information Pooling algorithm, however, incorporates a subroutine based on the “optimism in the face of uncertainty” principle, pooling information across different contexts. This subroutine is a key contribution that allows us to achieve tight regret rates on the order of $T^{2/3}$.
>
> **2)** This is undoubtedly an interesting question, which we leave as a direction for future research. Deriving meaningful lower or upper bounds for this setting will likely require entirely new techniques beyond those introduced in this paper or in previous works. Indeed, lower-bound constructions based on instances used in previous pricing or bilateral trade works have proven to be of limited use in this setting.

---

### Official Review · Reviewer_Dy9f · 2024-11-10

**Soundness:** 4
**Presentation:** 2
**Contribution:** 3
**Rating:** 8
**Confidence:** 3

**Summary:**

This paper tackles the bilateral trade problem in an online setting where there is an additional context present. At each time step $t$, a buyer and a seller arrive with private values $b_t$ and costs $s_t$. The algorithm must provide a pair of prices $(p,q)$ such that a sale happens if the buyer's value $b_t$ is less than $q$ and the seller's value is above $p$. The Gain from this trade assuming a sale happens is $(b_t - s_t)$.  The authors consider the problem of maximizing the gains from trade assuming either two bit feedback where we find out if $\mathbb{1}[b_t \leq q] $ and if $\mathbb{1}[s_t \geq p]$. They also consider the model where we have only one bit feedback where we know the product of these two bits of feedback. The original problem was already studied by Cesa-bianchi et al. This problem considers the setting where the buyer and seller have a hidden vector of preferences $\theta_b, \theta_s$ and their private values are generated from a shared context $b_t = x_t^T \theta_b$ and $s_t = x_t^T \theta_s$ . They consider where there may be some noise that is added as well as the budget balanced setting where the prices offered to both parties must be the same ($p=q$).


The main results:

1) In the two feedback setting with strong budget balance $p=q$ at each time step when there is no noise in the setting. Here the authors use a natural modification of the feature based toolification.
2) In the two bit feedback model with noise, where the noise is i.i.d coming from distributions with bounded support and densities. Finally they devise an algorithm following the explore-or-commit framework where the authors decompose the gain in terms
3) They also study the one bit feedback problem where you only find out if a sale happens or not. To get good bounds for this model they assume they have a good regret for the strongly-balanced two bit feedback and then use that in a black box manner. However the new bounds only have a global budget balanced guarantee.

**Strengths:**

The authors propose a very reasonable contextual model of online bilateral trade. I find the new model to be well motivated and combines two natural areas of study namely bilateral trade and online contextual regret minimization. The algorithms themselves seem interesting and are fairly natural.

The reduction from the two bit strong budget balanced case to the one bit global budget balanced case is perhaps the most interesting to me. Essentially it is a general recipe where by one can exploit explore-or-commit algorithms and perform the explorations in such a way that we can always get feedback about either the buyer or seller. However, we may lose regret compared to other party, and thus we need to ensure that there is sufficient budget to do this. This is done by measuring the average profit the 2 bit algorithm can learn and then appropriately setting the parameters to balance out the findings.

**Weaknesses:**

Although the algorithms are natural and interesting, I am unable to distinguish where the new ideas are and how much of the paper is using known tools to a new setting. I would appreciate more explanation on what the new ideas are in both the two bit setting and the one-bit setting. \o

**Questions:**

1) I am wondering why the stronger bounds from Contextual Search (Liu et al) were not used in place of the Feature Based Pricing. It seems many of the ideas would carry over and you would achieve improved regret guarantees.

2) It would be useful to know what new ideas are introduced for noisy setting  and how much was already known in other settings.

---

> ### Author Response · Authors · 2024-11-21
>
> We thank the reviewer for the positive comments about the paper.
>
> **1)** The Contextual Search algorithm is specifically designed for the dynamic pricing problem, since it exploits the structure of the problem to reduce the width of uncertainty sets and to maximize rewards. In particular, in the context of pricing, setting a price near the lower bound of the confidence interval increases the likelihood of a transaction, while limiting the reduction in reward to at most the width of the interval.
>
> The tradeoff in the bilateral trade problem is fundamentally different. In particular, any price within the range between the buyer’s and seller’s valuations yields the same reward. When the uncertainty sets for the buyer’s and seller’s valuations overlap, the optimal price clearly lies within this intersection. However, selecting a price closer to the lowest or highest endpoint of the intersection, or its midpoint, does not a priori improve the reward or increase the likelihood of a trade. Moreover, the loss from selecting a price that fails to result in a transaction, or the gain from choosing an appropriate price, is not directly tied to the width of the intersection.
>
> As a result, it is not obvious at all how Contextual Search algorithm techniques could be reconciled with bilateral trade, as they are inherently of different nature.
>
> **2)** We identify two key conceptual contributions which we present in this paper:
> 1. Our scouting algorithm introduces the key idea of sharing information across different contexts. This new approach, which has never been explored in prior work on the bilateral trade problem, is crucial to achieve the minimax optimal rate of $T^{2/3}$.
> 2. Our second main contribution is a general reduction scheme that allows us to apply ETC-like algorithm designed for two-bit feedbacks in the one-bit feedback setting. This reduction is new, and can be leveraged beyond the contextual setting to balance information gains against budget constraints.
>
> Along the way, we also introduce several new results that build on techniques previously used in the pricing and bilateral trade literature but require significant extensions due to the complexity of our setup. For example, Lemma 1 from Cesa-Bianchi et al. (2021) is extended to be able to handle the contextual version of the problem. Moreover, constructing unbiased estimators of linear expected valuations requires adapting techniques from noisy dynamic pricing to settings where features are chosen adversarially. These extensions could have broader applications beyond bilateral trade, for studying dynamic pricing problems in more challenging environments than those typically considered.

---

> > ### Comment · Reviewer_Dy9f · 2024-11-26
> >
> > Thank you for your response. I am happy to keep my score as is for the paper.

---

### Meta-Review · Area_Chair_zSd9 · 2024-12-21

**Metareview:**

The reviewers all appreciate the new scouting algorithm and the minimax optimal regret it achieves. A good paper overall.

**Additional Comments On Reviewer Discussion:**

NA

---

### Decision · Program_Chairs · 2025-01-22

Accept (Poster)